# Nonuniform and pathway-specific laminar processing of spatial frequencies in the primary visual cortex of primates

Tian Wang[1,2], Weifeng Dai[1], Yujie Wu[1], Yang Li[1], Yi Yang [1], Yange Zhang[1], Tingting Zhou[1], Xiaowen Sun[1], Gang Wang[3], Liang Li[3], Fei Dou[1,2] & Dajun Xing [1] ✉

The neocortex comprises six cortical layers that play a crucial role in information processing; however, it remains unclear whether laminar processing is consistent across all regions within a single cortex. In this study, we demonstrate diverse laminar response patterns in the primary visual cortex (V1) of three male macaque monkeys when exposed to visual stimuli at different spatial frequencies (SFs). These response patterns can be categorized into two groups. One group exhibit suppressed responses in the output layers for all SFs, while the other type shows amplified responses specifically at high SFs. Further analysis suggests that both magnocellular (M) and parvocellular (P) pathways contribute to the suppressive effect through feedforward mechanisms, whereas amplification is specific to local recurrent mechanisms within the parvocellular pathway. These findings highlight the non-uniform distribution of neural mechanisms involved in laminar processing and emphasize how pathway-specific amplification selectively enhances representations of high-SF information in primate V1.

The mammalian neocortex has six cortical layers with unique inputs, outputs and intracortical connections[1-4]. Layers 4 and 6 of the cortex, as input layers, receive feedforward information from other brain regions and transmit it to other layers in the same cortex. With intracortical processing, the output signals are sent to other cortical areas from cells in the output layers (layers 2/3 and 5). Although laminar processing is thought to be important in the sensory cortices of different species[3,5,6], the neural computations used to process information from multiple neuronal sources remain largely unknown.

In the primate visual system, the magnocellular (M) and parvocellular (P) pathways comprise two distinct functional streams for visual information[7-11]. The M pathway forms the basis for motion and depth-related processing[12-14], while the P pathway is a unique structure in primates that enables the visual perception of color and high spatial resolutions[15-18]. Anatomical studies have suggested that the information contained in the M and P pathways remains separate from the

retina to the lateral geniculate nucleus and is reorganized by the laminar circuitry among the cortical layers in the primary visual cortex (V1)[1,7,19]. In the geniculate recipient layers of V1 (the input layer, layer 4 C), the M and P pathways are still separate. The upper region of layer 4 C (L4Cα) receives projections from the M pathway, while the lower region (L4Cβ) receives projections from the P pathway[20,21]. Layer 4Cα conveys information in the M pathway mainly to layer 4B and partly to layers 2/3, but layer 4Cβ mainly conveys information in the P pathway to layers 2/3[22,23]. The distinct anatomical connections across the cortical layers in the M and P pathways indicate that the laminar processing strategy within V1 is pathway-specific.

One of the functional differences between the P and M pathways concerns their spatial resolutions. The P cells of the retina and geniculate nucleus have smaller receptive fields and prefer higher spatial frequencies (SFs) than M cells[24-29]. The spatial resolution difference between L4Cα and L4Cβ is maintained in the input layer of V1[30,31].

[1]State Key Laboratory of Cognitive Neuroscience and Learning and IDG/McGovern Institute for Brain Research, Beijing Normal University, Beijing 100875, China. [2]College of Life Sciences, Beijing Normal University, Beijing 100875, China. [3]Beijing Institute of Basic Medical Sciences, Beijing 100005, China. ✉e-mail: dajun_xing@bnu.edu.cn

Previous studies using stimuli with low/median SFs found suppressed laminar processing in V1, including a reduced response to the surface of an object[32], a nonpreferred orientation[33,34] and a large stimulus[35] in the output layer compared with that of the input layer. However, because low/median SF stimuli activate both the M and P pathways, it is difficult to examine the two pathways separately. Whether the laminar processing mechanism related to the P pathway is different from that related to the M pathway is still an open question.

Our study aimed to elucidate the pathway-specific laminar processing strategies in the M and P pathways. To this end, we used rapidly flashing grating patches presented at different orientations and different SFs and simultaneously recorded the spiking activity occurring in all V1 layers of awake macaques. We found two types of laminar processing (cross-layer suppression vs. amplification) under different SFs. Cross-layer amplification was correlated with high-SF processing, while suppressive patterns were found for all SFs. A further analysis revealed that distinct laminar processing mechanisms could be activated by small stimuli from the local circuitry within the column. A three-component model with two feedforward mechanisms (referred to as the M and P pathways) was found to prefer different SFs, and one recurrent mechanism predominantly in the P pathway was found to prefer high SFs, which could explain our above findings. Our results suggest that the M and P pathway-specific laminar processing mechanisms originate in local circuitries and play different functional roles in coding SF information in primate V1.

## Results

We recorded multiunit activity (MUA) and local field potentials (LFPs) in the V1s of three awake macaque monkeys (59 probe placements in total; DQ: 16 probe placements; DK: 26 probe placements; QQ: 17 probe placements) with linear multielectrode arrays (V-probes) (Fig. 1a). The probes were inserted perpendicular to the cortical surface to record responses from all layers within a column of V1. The receptive fields (RFs) overlapped across the channels of each placed probe (Fig. 1b). During each recording trial, the monkeys were trained to fixate on a spatial range (radius = 1°) around a small dot (radius = 0.1°) in the center of a screen, while a series of grating patches (4–8°) was briefly presented (for 20 ms) at different orientations to fully cover the RFs of the recorded sites (Fig. 1c). The spatial frequency (SF) of the grating patches was randomly chosen for each trial but fixed within each trial (Fig. 1c). We cross-correlated (also called reverse correlation or spike-triggered average; see Methods) the neural activity (MUA and LFPs) with stimuli and calculated the dynamic responses produced for different spatial frequencies.

For each probe placement, we assigned the relative cortical depth of each of the 24 probe channels based on a current source density (CSD) analysis of the visually evoked LFPs and the stimulus-driven MUA pattern (see Methods for details). The earliest CSD sink observed for low SFs was located at the input layer of the M pathway (L4Cα) (Fig. 1d); this is a typical pattern that has been shown in previous studies[34,36–40]. The CSD pattern for high SFs (10 cycles/degree) exhibited a stronger early sink in the input layer of the P pathway (L4Cβ) than in L4Cα (Fig. 1d), suggesting that higher SFs mainly activate the P pathway. SF-dependent CSD patterns were associated with the functional properties of the M and P pathways, supporting the precise assignment of cortical layers (see Supplementary Fig. 1 for the results of individual animals).

### Cortical suppression and amplification lead to diverse SF-dependent laminar patterns in V1

Based on the aligned cortical depth of the channels in each probe placement, we constructed the laminar response patterns of spike activities for different SFs (see Methods for details). We observed diverse laminar patterns at different spatial frequencies among all recording sessions (n = 59), which could be categorized into two types.

Figure 1e shows two examples derived from the same animal (DK). For the first type of laminar pattern (upper panels), the response was highest in the input layers (L4C) and was suppressed in the output layers (L2/3). For the second type of laminar pattern (lower panels), the output layers yielded stronger responses, especially to high SFs. Among all probe placements, the differences between the responses of the output and input layers (the response of the output layer minus the response of the input layer) under high-SF conditions exhibited a distribution with two peaks, which was significantly different from the unimodal distribution (Fig. 1f; n = 59; calibrated Hartigan's dip test, p = 0.02). The bimodal distribution of the response differences suggested that two cortical mechanisms (Fig. 1g) govern the diverse laminar response patterns induced under high SF conditions. The laminar patterns dominated by cortical suppression under high SF conditions generated a peak (the peak below 0 in Fig. 1f) in the distribution, and those dominated by cortical amplification generated the other peak (the peak greater than 0 in Fig. 1f).

For simplicity, we use the term "cross-layer" to describe the change in the response strength/property from the input layer to the output layer (between L2/3 and L4C) in this study. "Cross-layer amplification" indicates that L2/3 had a stronger response than L4C, and "cross-layer suppression" indicates that L2/3 had a weaker response than L4C. We further defined an output/input activation index as the ratio between the responses of the output layer (L2/3) and the input layer (L4C) for each probe placement under each SF condition. An index less than 1 for a probe placement under a SF condition indicates that cross-layer suppression (Fig. 1g) dominated the laminar response pattern induced by the SF, and an index greater than 1 represents that cross-layer amplification (Fig. 1g) dominated the laminar response pattern. Most probe placements showed cross-layer suppression at low SFs (57/59), but both cross-layer suppression and amplification were widely observed at high SFs (Fig. 1h). The output/input activation indices produced at high SFs were significantly greater than those induced at low SFs (Fig. 1i; p < 10⁻⁶; signed-rank test).

To test which data points (probe placements) had significant amplifications under high SF conditions, we compared the output/input activation index of each probe placement at a high SF with the output/input activation distribution produced at a low SF (upper panel of Fig. 1g). Among all probe placements, 29 penetrations had output/input activations significantly greater than 1 at a high SF and exhibited the mean suppression level at a low SF (black dots in Fig. 1i; n = 29, p < 0.05; bootstrap method: see Methods); 30 probe placements had output/input activations not significantly greater than 1 at a high SF and exhibited the mean suppression level at a low SF (gray dots in Fig. 1i; n = 30, p > 0.05; bootstrap method: see Methods). We categorized the 29 probe placements with significant cross-layer amplification in their laminar response patterns into a group (group 2) and the other 30 probe placements with cross-layer suppression into another group (group 1) for further investigation. The two groups also exhibited a difference in a k-means analysis (see Supplementary Fig. 2) of their laminar response patterns.

The observation of diverse laminar processes for different SFs (Fig. 2a) was also confirmed by averaging the patterns of the responses produced for similar SFs within each group (Fig. 2b; see Supplementary Fig. 3 for the results of individual animals). Under low SF conditions, both groups had strong responses in the upper input layer (L4Cα) and showed suppressive effects in the output layer relative to the input layer. Under high SF conditions, the responses in the lower input layer (L4Cβ) were greater than those in the upper input layer (L4Cα); one group exhibited cross-layer suppression (upper panels of Fig. 2b), and the other group exhibited cross-layer amplification (lower panels of Fig. 2b).

Most recording sites in L2/3 demonstrated cross-layer suppression at low SFs, while cross-amplification was mainly observed at high

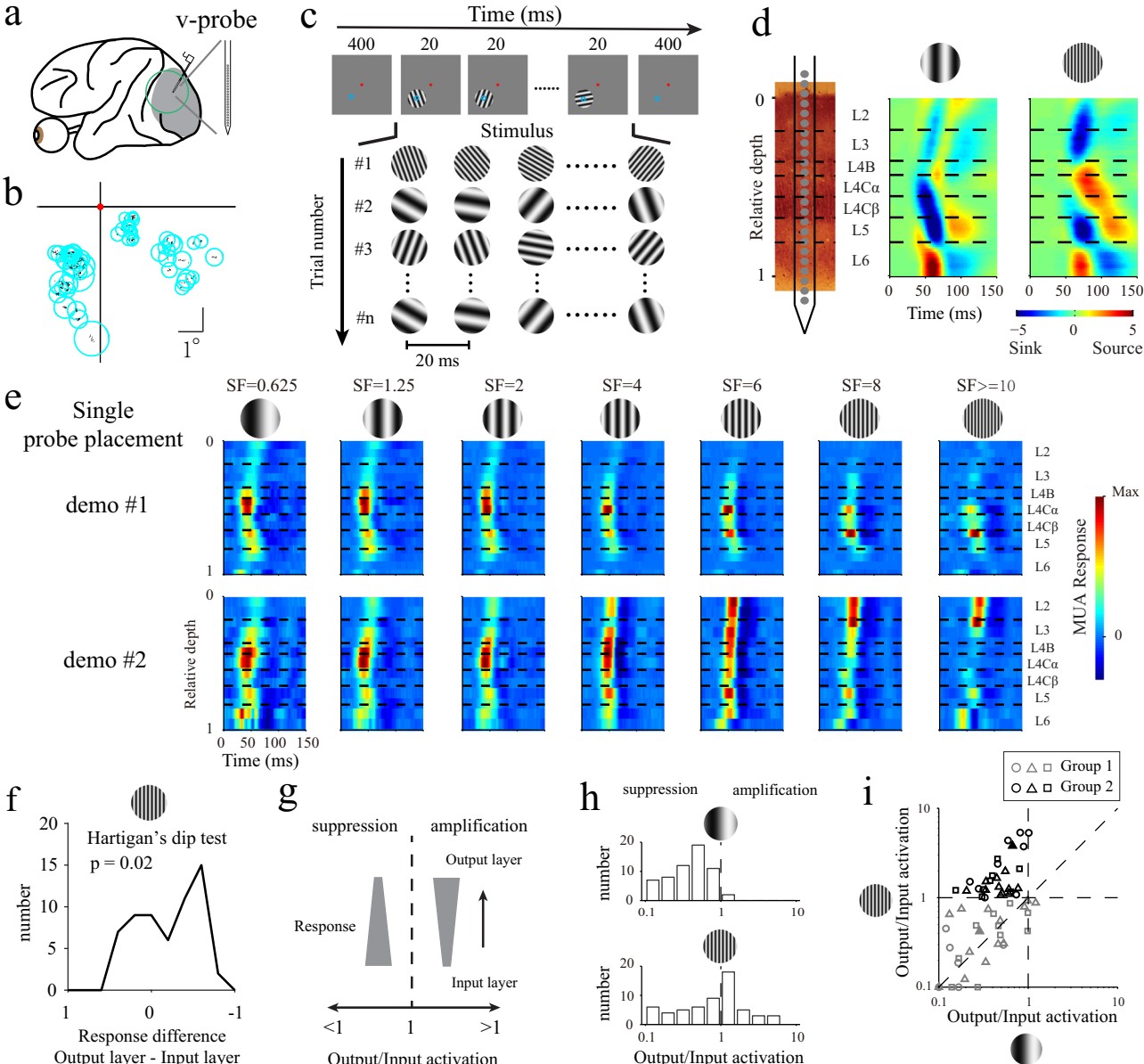

**Fig. 1 | Diverse laminar processing mechanism in the primate V1. a** Spiking activity and local field potential were recorded with a V-probe (Plexon, 24 channels, interchannel spacing of 100 μm). The gray region represents V1. **b** Spatial distributions for the RF centers (dots) of the recorded sites derived from all probe placements. The circles represent the mean locations (centers) and sizes (radii) of the RFs determined from each probe placement. **c** Stimulus paradigm. A series of grating patches with different orientations were presented on a gray screen. Each grating patch was presented for 20 ms. The red dots and blue circles represent the fixation point locations and the RFs of the recorded sites, respectively. **d** Laminar recording and assignment results. Left panel: The linear array was positioned vertically through the full depth of V1. Middle and right panels: Laminar CSD patterns for low (1.25 cycles/degree) and high SFs (10 cycles/degree) averaged over all probe placements. The horizontal dashed lines indicate the laminar boundaries in V1. **e** Laminar response patterns of the MUAs obtained for seven SFs from two individual probe placements. Grating stimuli with different SFs are shown at the top of the figure. The strength of MUA response is indicated by its color. Each SF pattern was normalized by dividing it by its maximum value. **f** Distribution of the response differences between the output and input layers (calibrated Hartigan's dip test for bimodal distribution, $n = 59$ probe placements, $p = 0.02$). **g** Schematic for quantifying the laminar processing mechanism. The width of each trapezoid represents the response strength. Output/input activation was defined by the ratio of the responses between the output layers and the input layers at the recording sites. **h** Distributions of the output/input activations for a low SF (upper panel) and a high SF (lower panel). The output/input activations were averaged from all sites within L2/3 for each probe placement ($n = 59$). **i** Relationship between the output/input activations calculated from the low SF and high SF conditions. Gray represents group 1, and black represents group 2. The different shapes represent the three animals (DQ: circles; DK: triangles; QQ: squares). Filled triangles is calculated from data of probe placements shown in (**e**). Source data are provided as a Source Data file.

SFs for group 2 (Fig. 2c). The output/input activation patterns between the sites in the two groups were significantly different under high SF conditions (Fig. 2c; sites from L2/3; 0.34 ± 0.07 for group 1, $n = 100$; 1.180 ± 0.25 for group 2, $n = 126$; rank-sum test: $p < 10^{-11}$; for SF stimuli higher than 10 cycles/degree). This difference between the two groups was not due to bias in the neural signal quality (as quantified by the SER; see Methods). The SER values were not significantly different between the sites of the two groups in the input layer for all SFs (Supplementary Figs. 4a–d). In the output layer, differences were found only for SFs larger than 2 cycles/degree (Supplementary Fig.4e–h). Both groups had high SER values under low SF conditions (Supplementary Fig. 4a, e), which suggested that the neurons at the

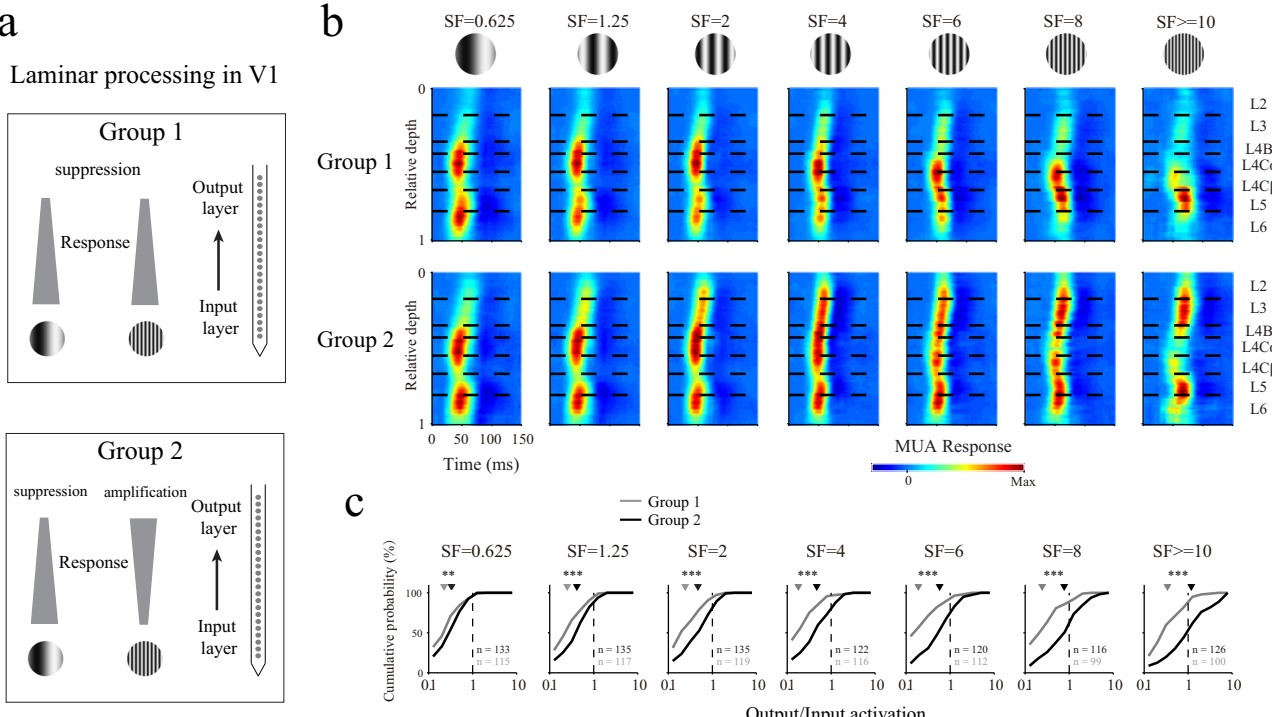

**Fig. 2 | Laminar processing is dependent on the SF. a** Schematic of two groups with different laminar processing patterns at high SFs. **b** Population-averaged laminar patterns for different SFs. Grating stimuli with different SFs are shown at the top of the figure. The two groups of columns are presented separately (the upper panels show group 1, and the lower panels show group 2). The strength of each MUA response is indicated by its color. The length of the sliding window for averaging across the depth dimension was 0.1 (relative depth). The horizontal black dashed lines represent the laminar boundaries. Each pattern of the SF response was normalized by dividing it by its maximum value. **c** Cumulative probability distributions of the output/input activations for L2/3. For each probe placement, the responses of the input layers from all the L4Cα and L4Cβ sites were averaged. The responses were averaged from 0 to 120 ms after the onset of the stimulus. For each SF condition, sites with responses higher than 0 were included (n is the number of valid sites). Gray represents group 1, and black represents group 2. **p < 0.01, ***p < 0.001, two-sided rank-sum test (p value for seven SF conditions: $p = 0.005$, $p < 10^{-3}$, $p < 10^{-6}$, $p < 10^{-8}$, $p < 10^{-10}$, $p < 10^{-10}$, $p < 10^{-11}$). The average values are indicated by triangles. Source data are provided as a Source Data file.

recording sites located in the two groups were effectively activated by visual stimuli.

To exclude the possibility that the diverse laminar processing patterns we categorized into two groups were caused by changes in the response properties across recording sessions, we simultaneously recorded from two probe placements with two V-probes for 10 sessions (Supplementary Fig. 5a; two animals: DK and QQ). The distinct distributions of the RF centers produced under the two probe placements indicated that they were located in different columns (Supplementary Fig. 5b). Diverse laminar processing patterns for different SFs were also observed with simultaneous recordings obtained from various probe placements, not only for individual recording sessions (Supplementary Fig. 5c) but also for population-averaged patterns (Supplementary Figs. 5d, e); this suggested that the diverse laminar processing patterns were caused by functional differences between the two groups rather than changes in the response properties across time[41].

It is well known that sensitivity to a spatial frequency varies strikingly across retinotopic eccentricity variations[30,42,43]. To test the influence of eccentricities, we further analyzed the eccentricities of the recording sections. First, we measured the impact of eccentricity on the cutoff SFs in both the input and output layers (Supplementary Fig. 6a). There were significant correlations between the eccentricity and cutoff SFs for both L4Cα ($r = -0.5$, $p < 10^{-4}$) and L4Cβ ($r = -0.38$, $p = 0.003$), and the correlation observed for L2/3 was also significant but relatively weak ($r = -0.29$, $p = 0.025$). Then, we selected four SF conditions based on the cutoff SF in L4Cβ for each probe placement (0.25, 0.5, 1 and 2 octaves relative to the 50% cutoff SF in L4Cβ; two examples are shown in Supplementary Fig. 6b). Because the four SF conditions were based on the cutoff SF in L4Cβ for each probe

penetration, the effect of eccentricity was removed/minimized. The average laminar patterns obtained for the two groups under the four selected SF conditions (Supplementary Fig. 6c) were similar to those shown in Fig. 2b, which suggested that the cross-layer suppression and amplification effects induced under high SF conditions were not caused by the effects of eccentricities.

Overall, we found that the diverse laminar processing patterns obtained under different SFs in primate V1 could be divided into two types (cross-layer suppression and cross-layer amplification). The cross-layer amplification patterns were mainly evoked by high SFs for a specific group of cortical locations (columns), while suppressive patterns were found for all SFs (from low to high). The diverse laminar response patterns observed at different cortical locations in our study are consistent with the different functional columns in V1 shown in previous studies, such as the blob and interblob[30,44,45]. We further discuss the relationship between the diverse laminar processes in our results and the cytochrome oxidase architecture in the discussion section.

## Cortical amplification enhanced the representation of high-SF information

The two groups of recordings also exhibited different degrees of the selectivity to the SFs in their output layers, in addition to having different laminar response patterns. Figure 3a shows the SF tuning results of individual sites in the output layer (L3), with greater SF preference than that exhibited by L4Cβ for the same probe placement. The interlaminar increase in SF preference from the input layer to the output layer was large for group 2 (Fig. 3b; left panel, another example site) but moderate or small for group 1 (Fig. 3b; right panel). We quantified the distribution of the SF preferences and their changes

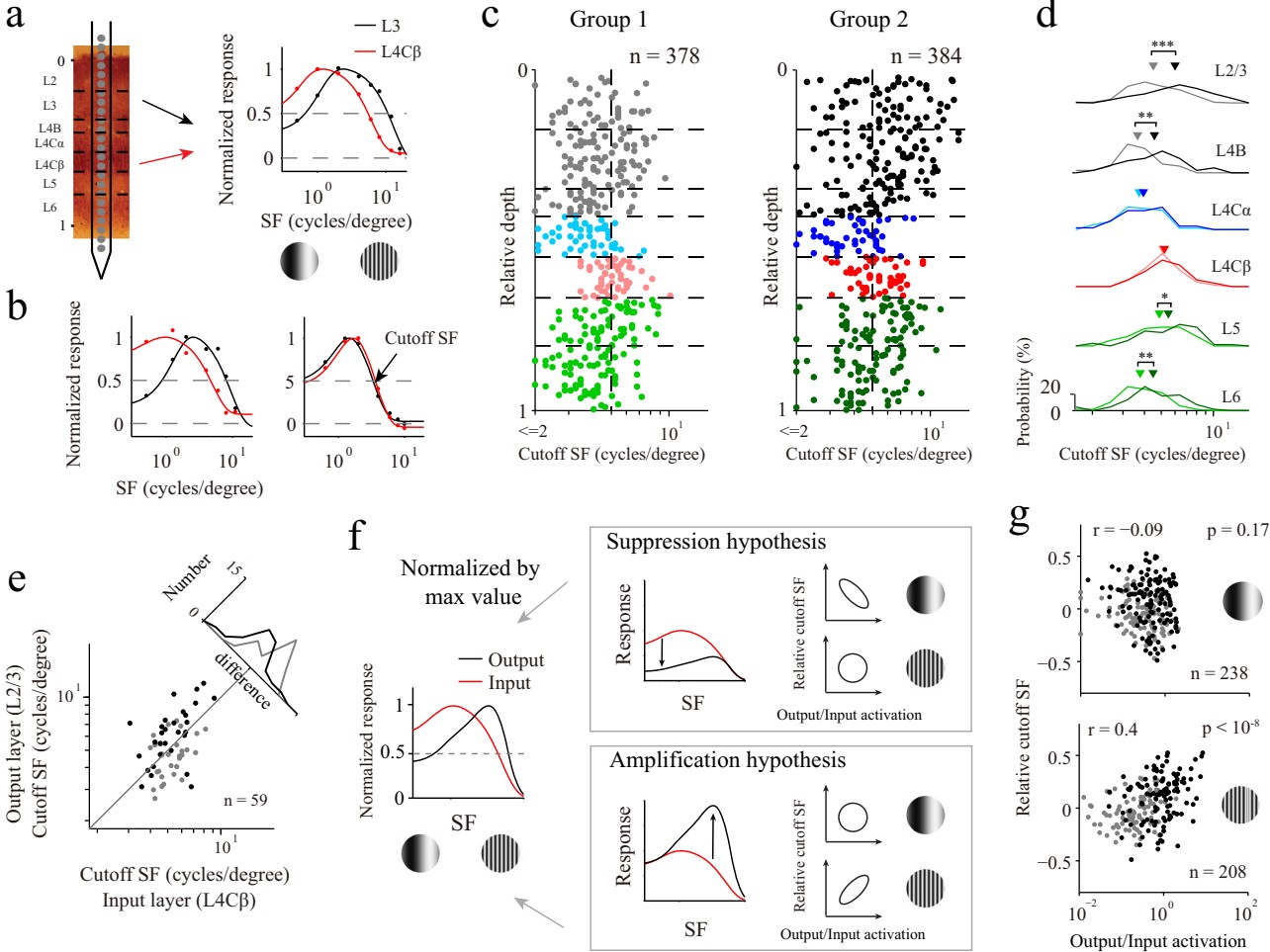

**Fig. 3 | Laminar processing patterns correlated with SF representations. a** SF tuning curves (normalized by dividing them by the peak values) obtained for two example recording sites from the same probe placement. The colors of the lines represent different layers. Red and black curves indicate different levels of data fitting (dots). **b** Two additional example recording sites. The black arrow indicates the cutoff SF. **c** Laminar distributions of the cutoff SFs for the two groups. The different colors represent the different layers. **d** Histogram of the cutoff SFs for different groups and cortical layers. The average values are indicated by triangles. Two-sided rank-sum test is used ($p < 10^{-7}$ for L2/3; $p = 0.009$ for L4B; $p = 0.31$ for L4Cα; $p = 0.54$ for L4Cβ; $p = 0.04$ for L5; $p = 0.002$ for L6). **e** Relationship between the cutoff SF of the input layer (L4Cβ) and the cutoff SF of the output layers (L2/3) ($n = 59$ probe placements). Each dot represents data averaged from all sites for the same probe placement within each layer. Gray represents group 1, and black represents group 2. The histogram shows the difference between the cutoff SFs of L2/3 and L4Cβ. **f** Illustration of two hypotheses for the SF preference shift. The red line represents the input layer, and the black line represents the output layer. The direction of the black arrow represents the response change from the input layer to the output layer. The ellipse indicates a high correlation. A circle indicates a plot if no correlation was observed. **g** Relationship between the output/input activation and the relative cutoff SF. The data for each dot were calculated from one pair of sites for L2/3 and L4Cβ at the same probe placement ($n$ is the number of valid pairs, and $r$ is the Pearson's correlation coefficient; $p = 0.17$ for low SF, $p < 10^{-8}$ for high SF). The methods for the selection of the low- and high-SF conditions are described in the Methods section. Source data are provided as a Source Data file.

across the V1 layers by calculating the cutoff SFs (the highest SF reaching 50% of the peak response) for individual sites (Fig. 3b; indicated by the black arrow in the right panel). Most L2/3 sites with cutoff SFs greater than those of L4 were derived from group 2 (Fig. 3c; right vs. left panels). Interestingly, the cutoff SFs of the two groups were significantly different in the output layers and deep layers but not in the input layers (Fig. 3d; rank-sum test: $p < 10^{-7}$ for L2/3; $p = 0.009$ for L4B; $p = 0.04$ for L5; $p = 0.002$ for L6; $p = 0.31$ for L4Cα; $p = 0.54$ for L4Cβ). This suggests that the SF preference increase was caused by laminar processing within V1 rather than a precortical mechanism influencing transmission from the LGN to the input layers of the two groups. Although the cutoff SFs of the two groups were significantly different, the overall distribution of all cutoff SFs in the V1 output layers was not significantly different from a unimodal distribution (calibrated Hartigan's dip test, $p = 0.19$). The continuously distributed SF sensitivity observed in our study is consistent with the findings of previous studies[46,47].

To further reveal the interlaminar SF preference changes between the two groups, we compared the cutoff SFs of the output layer (L2/3) with those of the input layer (L4Cβ) for each probe placement ($n = 59$). For the probe placements classified as belonging to group 2, the cutoff SFs mostly increased (Fig. 3e, black dots; $n = 29$; $6.49 \pm 0.44$ for L2/3; $5.42 \pm 0.27$ for L4Cβ; signed-rank test, $p = 0.003$). For the probe placements classified as belonging to group 1, the cutoff SFs mostly decreased (Fig. 3e, gray dots; $n = 30$; $4.75 \pm 0.23$ for L2/3; $5.32 \pm 0.19$ for L4Cβ; signed-rank test, $p = 0.029$). This suggests that the representations of high SF information differed between the two groups with different laminar response patterns. Furthermore, the cutoff SFs of L2/3 were significantly correlated with the output/input activations at high SFs (Supplemental Fig. 7a; $n = 59$, $r = 0.38$, $p = 0.003$), which indicates that neural mechanisms (suppression and amplification) not only lead to diverse laminar response patterns but also strongly affect the functional properties of V1 output layers.

The SF preference difference between the input layer and the output layer may be explained by two different mechanisms/hypotheses (Fig. 3f). The first hypothesis (the suppression hypothesis) is that the responses to low SFs are largely suppressed, but the responses to high SFs are less affected, so the SF preference in the output layer shifts to high SFs (Fig. 3f; upper panel). The other possible mechanism (the amplification hypothesis) is that the selective amplification of the responses to high SFs causes an increase in the SF preference (Fig. 3f; lower panel). These two hypotheses are associated with distinct expectations regarding the correlation between the response differences and cutoff SFs between the input layer and the output layer. For the suppression hypothesis, a significant correlation between the interlaminar differences between the responses to low SFs and cutoff SFs is expected. In contrast, according to the amplification hypothesis, a significant correlation between the interlaminar changes exhibited in response to high SFs and the cutoff SFs is expected.

To test which hypothesis best explained our findings, we generated scatter plots of the response changes (defined by the output/input activations) and the cutoff SF changes (relative cutoff SF; see Methods for details) (Fig. 3g; each dot was calculated from one pair of sites from L2/3 and L4Cβ at the same probe placement). A significant correlation was found for high SFs but not for low SFs, supporting the amplification hypothesis ($n = 238$ pairs for low SFs, $r = -0.09$, $p = 0.17$; $n = 208$ pairs for high SFs, $r = 0.4$, $p < 10^{-8}$). The results suggest that the functions of the two types of laminar processing are different, and cross-layer amplification increased the response to high SFs and enhanced the selectivity to high SFs in the output layer of V1.

### Local circuitry is sufficient for generating diverse laminar processing patterns in V1

The results in the previous sections demonstrated the presence of two types of laminar processing for SF stimuli. Previous studies have suggested that large stimuli can effectively activate horizontal/feedback connections[37,38,48,49], which facilitate the integration of high-SF information such as contour lines in a large visual field[50–52]. A related question is whether laminar processing differences (especially cross-layer amplification at high SFs) are due to local circuitry (feedforward and local recurrent mechanisms) or whether interlaminar suppression and amplification utilize global circuitry such as horizontal/feedback connections (Fig. 4a). To isolate the effects of local circuitry from the horizontal/feedback connections, we also used small grating patches (0.8–1.2°) to evoke a laminar response in V1 ($n = 32$ probe placements), which is considered to mainly involve activating the local circuitry within the column (Fig. 4a).

The laminar patterns produced in response to small grating patches were similar to those yielded in response to large stimuli not only for the results of individual probe placements (Fig. 4b) but also for the results of population averages (Figs. 4c, 2b). The change between the output/input activations obtained with the SFs of the two groups persisted under the small stimulus condition (Fig. 4d; sites from L2/3; $0.232 \pm 0.070$ for group 1, $n = 42$; $1.274 \pm 0.877$ for group 2, $n = 73$, rank-sum test, $p < 10^{-8}$; at SF stimuli higher than 10 cycles/degree). The output/input activations between the small and large stimulus sizes were highly correlated, further suggesting that the activation of horizontal/feedback connections by a large stimulus did not change the overall laminar response pattern (Fig. 4e; $n = 118$ for sites in L2/3 in the high SF condition; $r = 0.75$, $p < 10^{-21}$; also see Supplementary Fig. 7b for the averaged output/input activation of each probe placement). There was no significant difference between the output/input activations produced for small and large stimuli (Fig. 4e; signed-rank test, $p = 0.74$). Our results suggest that both suppressive and amplified cross-layer processing originate in local circuitry and may not require global circuitries such as horizontal connections.

We also found some differences between the functional properties produced under the small and large stimulus conditions. The firing rate of the recording sites was significantly reduced for large stimuli; this is consistent with previous studies concerning surround suppression (Fig. 4f; for L2/3, high SF condition; group 1: $n = 49$, small size $9.40 \pm 0.93$, large size $7.32 \pm 0.48$, $p < 10^{-4}$; group 2: $n = 78$, small size $14.45 \pm 1.64$, large size $12.00 \pm 1.13$, $p < 10^{-3}$; signed-rank test). The cutoff SF was also significantly reduced for large stimuli (Fig. 4g; for L2/3; group 1: $n = 49$, small size $5.10 \pm 0.17$, large size $4.68 \pm 0.19$, $p = 0.004$; group 2: $n = 75$, small size $6.25 \pm 0.16$, large size $5.62 \pm 0.17$, $p = 0.002$; signed-rank test). The reduced firing rate and cutoff SF suggest that global circuitries do not amplify the responses to high SFs but mainly suppress the responses in V1.

### Recurrent connections within the V1 output layer regulate the cortical amplification of high SFs

The finding that the laminar processing suppression effect is generated by local circuitry is consistent with previous studies[33,34,53,54] concerning the neural mechanisms underlying the suppression of laminar processing (Figs. 2b and 4c). However, the neural mechanisms of cross-layer amplification for high SFs have not been explored. Two possible explanations for the cross-layer amplification in V1 are illustrated in Fig. 4a. One possibility is that selective amplification is caused by the feedforward connectivity from the input layers to the output layers (feedforward mechanism; middle panels of Fig. 4a). Another possibility is that cross-layer amplification is due to the recurrent connectivity within the output layers (local recurrent mechanism; right panels of Fig. 4a).

To test these two possible mechanisms for the cortical amplification of high-SF information, we estimated the connectivity strengths across the different cortical layers within V1 via a Granger causality analysis (GC; see Methods for details). For each probe placement, we calculated a connection matrix between the channels within V1. Feedforward connections were defined as connections from the input layer to the output layer, while recurrent connections were defined as connections from the output layer to the output layer (Fig. 5a). To obtain a connection pattern with a high resolution, we averaged the results obtained for all probe placements and SF conditions. Interestingly, we found different connectivity patterns for the two groups (Fig. 5b; small stimulus; 32 probe placements; $n = 14$ for group 1, $n = 18$ for group 2). For the group with cross-layer amplification, stronger recurrent connectivity was observed within L2/3 than that of the group dominated by suppression (Fig. 5c). A further statistical analysis revealed that the strength of the recurrent connections (from L2/3 to L2/3) was significantly greater in the group with amplification, but this pattern was not observed for feedforward connections (from L4C to L2/3) (Fig. 5d; $p = 0.01$ for recurrent, $p = 0.23$ for feedforward connections, rank-sum test). We further defined the weight of recurrent connections (normalized GC value) as the ratio of the GC value for recurrent connections to that for full connections (feedforward and recurrent connections). The maximum normalized GC value was observed for high SFs in the group with cross-layer amplification (Fig. 5e; Welch's ANOVA, F value = 4.54, $p = 0.02$). A further analysis revealed that the normalized GC value was significantly correlated with the output/input activation under the high SF condition (Fig. 5f; $r = 0.43$, $p = 0.01$, $n = 32$), which indicates that when the weight of the recurrent connections is high, the laminar response pattern is more likely to exhibit cross-layer amplification. The results obtained from the Granger analysis suggest that the neural mechanism for cross-layer amplification in response to high SFs is the selective activation of the local recurrent connections within the output layers.

### Laminar processes differ between the M and P pathways

Our findings concerning multiple laminar processes for SF information indicate that the neural computations of the M and P pathways are different because different SFs activate the M and P pathways differently in terms of both evoked responses (Figs. 2b, 4c) and SF

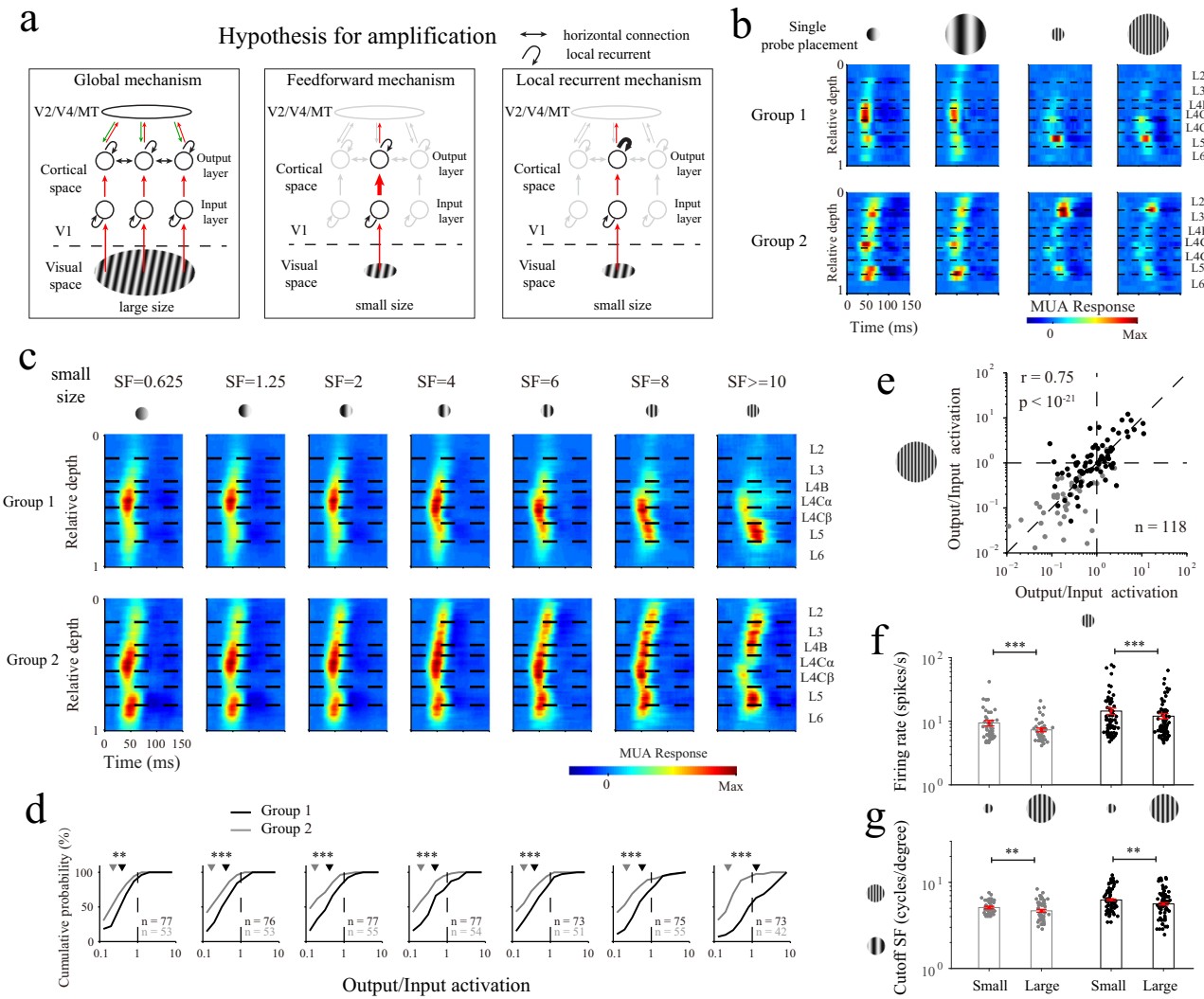

**Fig. 4 | Local circuitry is sufficient for generating multiple laminar processing patterns in V1. a** Schematic of the three amplification generation hypotheses. The circuitries in V1 include feedforward connections, local recurrent connections and global connections (horizontal connections and feedback connections). Red arrows indicate feedforward connections. The green arrows in the left panel indicate feedback connections from the higher visual cortex (V2/V4/MT). The connections not activated in the middle and right panels are represented in light gray. The thickness of an arrow represents the strength of the corresponding connection. **b** Laminar response patterns evoked by different stimulus sizes from two individual probe placements (the upper panels show group 1; the lower panels show group 2). The horizontal black dashed lines represent the laminar boundaries. Each SF pattern was normalized by dividing it by its maximum value. **c** Population-averaged laminar patterns evoked by small stimuli with different SFs. The strength of an MUA response is indicated by its color. The length of the sliding window for averaging across the depth dimension was 0.1 (relative depth). **d** Cumulative probability distributions of the output/input activations for L2/3. Gray represents group 1, and black represents group 2. Two-sided rank-sum test ($p$ value for seven SF conditions: $p = 0.003$, $p < 10^{-3}$, $p < 10^{-4}$, $p < 10^{-4}$, $p < 10^{-3}$, $p < 10^{-4}$, $p < 10^{-8}$). The average values are indicated by triangles. **e** Relationship between the output/input activations calculated from small and large stimuli under the high SF condition (for L2/3, $n = 118$; Pearson's correlation, $r = 0.75$, $p < 10^{-21}$). **f** Firing rates of L2/3 neurons for small and large stimuli under the high SF condition. Bars represent the mean values across populations ($\pm$ SEM), with individual data superimposed (dots). Gray represents group 1 ($n = 49$ sites), and black represents group 2 ($n = 78$ sites). Two-sided signed-rank test ($p < 10^{-4}$ for Group 1, $p < 10^{-3}$ for Group 2). **g** Similar to F but for cutoff SFs ($n = 49$ sites for group 1, $n = 75$ sites for group 2). Two-sided signed-rank test ($p = 0.004$ for Group 1, $p = 0.002$ for Group 2). ** $p < 0.01$, *** $p < 0.001$. Source data are provided as a Source Data file.

preferences (Fig. 3c, d). Our next goal was to dissect the mechanisms associated with the M and P pathways from the various laminar response patterns with a computational model. The model included three components corresponding to two feedforward components in the M and P pathways and one component for the recurrent mechanism (Fig. 6a; FF & Rec model). We assumed that the neural response to any SF in each V1 layer was a linear sum of the activations due to the use of two feedforward mechanisms and one recurrent mechanism. Neural activation due to the feedforward mechanism was generated by convoluting the neural responses in the input layers (L4Cα for the M pathway and L4Cβ for the P pathway) and weighted ($W_m$ and $W_p$) temporal kernels. Neural activation due to the local

recurrent component in each cortical layer was generated by a layer-specific temporal response multiplied by the recurrent weights ($W_{rec}$), which was dependent on the SF (Fig. 5e). The temporal kernels for feedforward activation and the time courses for recurrent activation were all modeled as log-normal functions (Eqs. 5 and 6), which were layer-dependent but fixed for all SF conditions. The weights for feedforward activation ($W_m$ and $W_p$) were layer-dependent but fixed for all SFs, and the weights ($W_{rec}$) for the local recurrent mechanism were dependent on both the layers and SFs. We optimized the model parameters by minimizing the mean squared error between the predicted and observed response dynamics of V1 (see Methods for the details of the model). Figure 6b shows a model prediction example

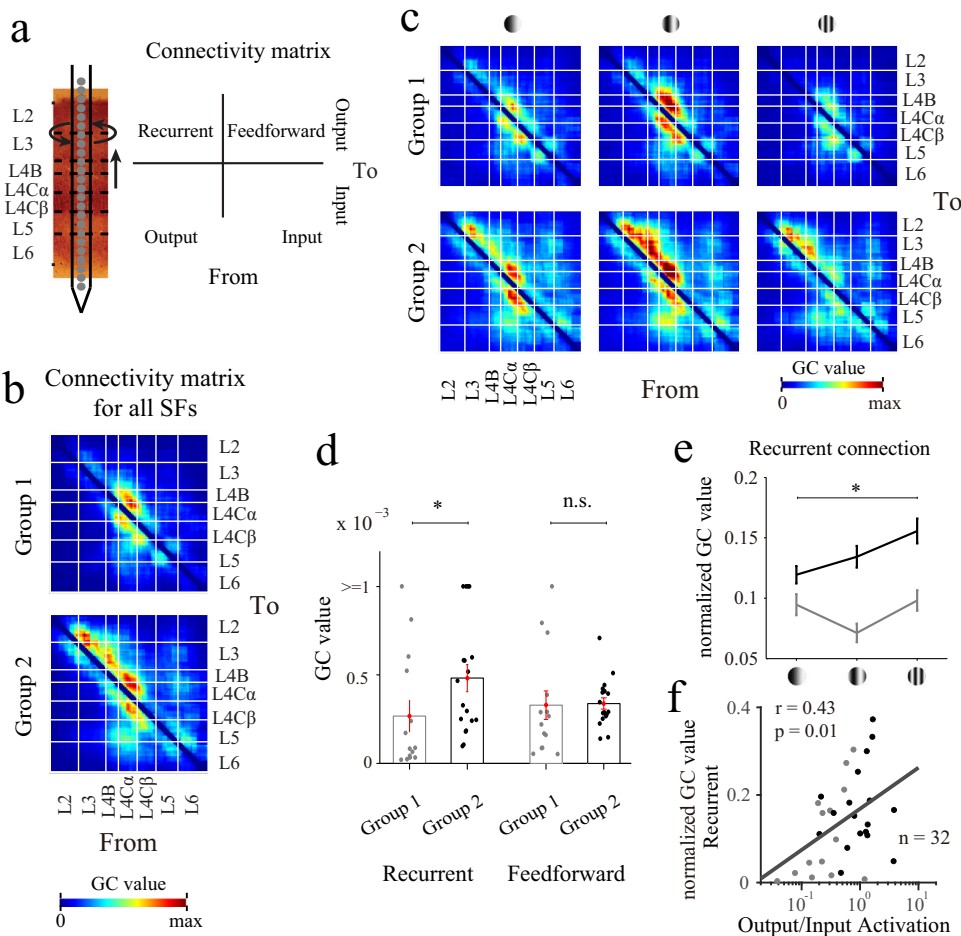

**Fig. 5 | Local recurrent connections within the output layer govern laminar processing. a** Schematic of the connectivity matrix with feedforward and recurrent connections. **b** The population-averaged connectivity within V1, averaged from all SFs. Thin lines indicate the laminar boundaries in V1. **c** Population-averaged connectivity levels for different SFs (three levels: low, medium and high; these are shown from the left panel to the right panel). The low SF condition was defined as SFs lower than 2 cycles/degree, the medium condition was defined as SFs ranging from 2 to 6 cycles/degree, and the high condition was defined as SFs greater than or equal to 6 cycles/degree. **d** GC values of the feedforward and recurrent connections averaged from all SFs. Bars represent the mean values across populations (±SEM), with individual data superimposed (dots). The strength of the recurrent connections was calculated based on the GC values from L2/3 to L2/3. The strength of the

feedforward connections was calculated according to the GC values from L4 to L2/3. Gray represents group 1 ($n = 14$ probe placements), and black represents group 2 ($n = 18$ probe placements). Two-sided signed-rank test ($p = 0.01$ for recurrent, $p = 0.23$ for feedforward connections). **e** Change in the normalized GC value with different SFs. The normalized GC value was defined as the ratio between the strengths of the recurrent connections and full connections (feedforward and recurrent). Welch's ANOVA, two-sided, $p = 0.02$. **f** Relationship between the output/input activation and the normalized GC value calculated from stimuli under the high SF condition ($n = 32$; Pearson's correlation, $r = 0.43$, $p = 0.01$). Linear regression (gray line) was also calculated for correlation measurements. *$p < 0.05$, n.s., not significant. Source data are provided as a Source Data file.

regarding the individual probe placement results obtained under high SF conditions.

To check whether the recurrent mechanism was necessary, we also tested an alternative model in which the V1 responses in different cortical layers were simply explained by two feedforward mechanisms (the FF model) of the M and P pathways. The goodness-of-fit of the FF model was significantly lower than that of the FF & Rec model (Fig. 6b, c; for L2/3, FF model: $0.72 \pm 0.01$; FF & Rec model: $0.81 \pm 0.01$; $N = 254$, signed-rank test, $p < 10^{-42}$). The good performance of our FF & Rec model suggests that the recurrent component was necessary for explaining the V1 response (see Supplementary Fig. 8 for population-averaged results).

The FF & Rec model enabled us to dissect the laminar activation patterns produced for the feedforward and recurrent mechanisms associated with the M and P pathways (Fig. 6d shows population-averaged laminar patterns of the three mechanisms for all SFs; see Supplementary Fig. 9 for each SF condition). The contributions of the three mechanisms to the laminar activation patterns varied with

different SF stimuli. The contribution of the feedforward mechanism in the M pathway was large when the stimuli had low SFs and small when the stimuli had high SFs. In contrast, the feedforward mechanism of the P pathway was engaged for high SFs (Fig. 6e). This finding is consistent with research showing that the M pathway prefers a low spatial frequency and that the P pathway prefers a high spatial frequency[18,24,30]. We further calculated the MP index, which is defined as the relative contribution of the feedforward components of the M and P pathways (see Methods for details). The MP index increased with the SF, with the M pathway dominating at low SFs (MP index less than 0) and the P pathway dominating at high SFs (MP index greater than 0). Importantly, the contribution of the recurrent mechanism increased with the SF in a similar manner to that of the MP index (Fig. 6f) and was significantly correlated with the activation of the P pathway (Fig. 6g; $n = 59$, $r = 0.39$, $p = 0.002$), suggesting that the local recurrent mechanism is also P pathway-specific.

To further elucidate the relationships between the three mechanisms (two feedforward mechanisms and one recurrent

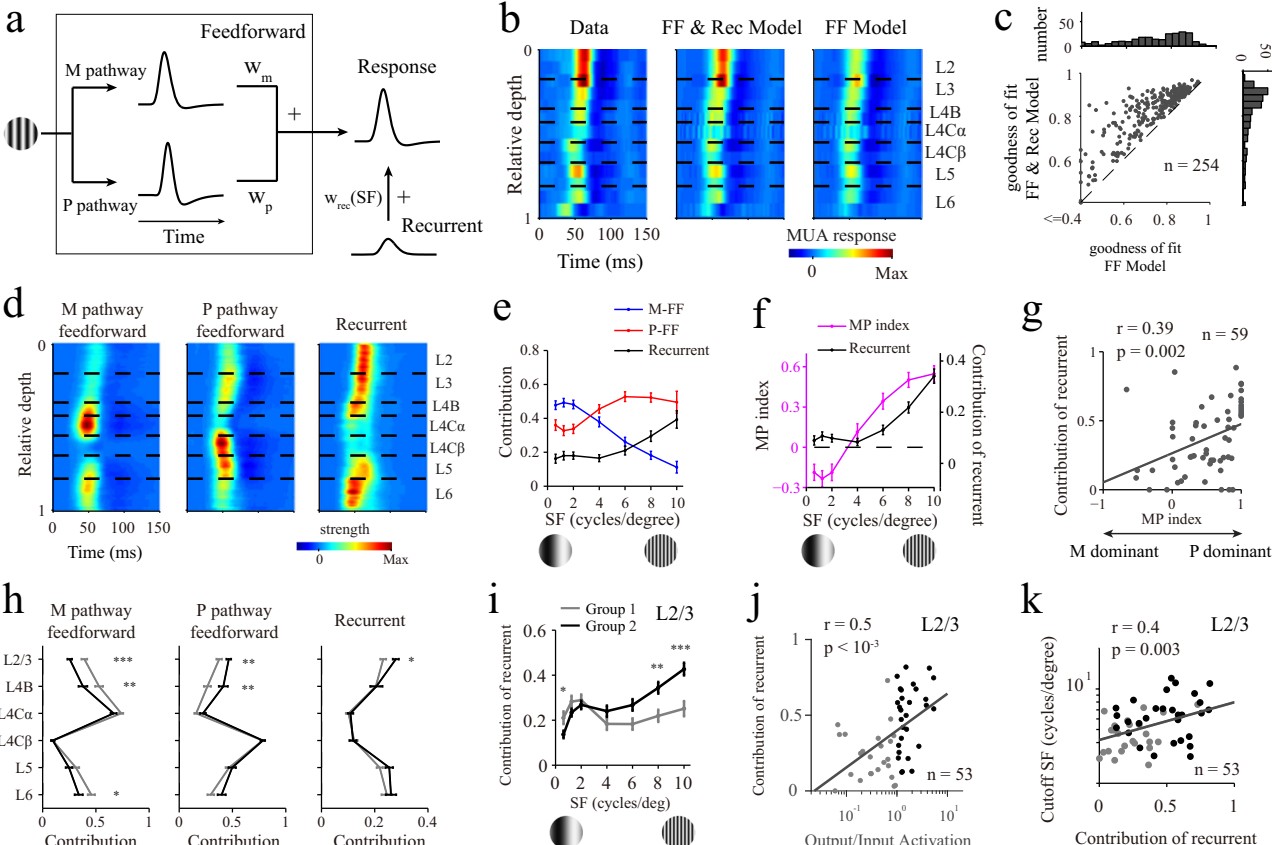

**Fig. 6 | A model with pathway-specific components explained the multiple laminar processing patterns. a** Illustration of a model for generating responses in V1. **b** An example of the model fitting results obtained for individual probe placements at an SF with 8 cycles/degree. Left panel: experimental data; middle panel: FF & Rec model; right panel: FF model. **c** Performance comparison for explaining the neuronal responses in the output layer between the FF model and the FF & Rec model. $n = 254$ sites from L2/3. **d** Population-averaged laminar patterns of the three components of the FF & Rec model. **e** Contributions of the three components to the laminar patterns produced at different SFs ($n = 59$ probe placements). **f** Changes in the relative contributions of the feedforward components derived from the M and P pathways (MP index) with different SFs ($n = 59$ probe placements). **g** Relationship between the MP index and the contribution of the recurrent component under the high SF condition (Pearson's correlation, $r = 0.39$, $p = 0.002$). **h** Contributions of the three components in different cortical layers. The line colors represent the two groups (two-sided rank-sum test; L2/3 to L6 of group 1: $n = 62, 35, 49, 53, 53, 41$ sites; L2/3 to L6 of group 2: $n = 109, 31, 46, 48, 46, 43$ sites; $p$ value for left panel, L2/3 to

L6, $p < 10^{-4}$, 0.008, 0.06, 0.92, 0.15, 0.03; $p$ value for middle panel, L2/3 to L6, $p = 0.006, 0.01, 0.17, 0.97, 0.44, 0.09$; $p$ value for right panel, L2/3 to L6, $p = 0.03$, 0.67, 0.63, 0.68, 0.09, 0.64). **i** The contribution of the recurrent component changed by the SF at L2/3 (two-sided rank-sum test; $n = 62$ sites for Group 1, $n = 109$ sites for Group 2; $p$ value for seven SF conditions: $p = 0.03$, $p = 0.12$, $p = 0.44$, $p = 0.27$, $p = 0.11$, $p = 0.004$, $p < 10^{-3}$). **j** Relationship between the output/input activation and the contribution of the recurrent component calculated from stimuli under the high SF condition. Each dot represents data acquired from one probe placement (Pearson's correlation, $n = 53$, $r = 0.5$, $p < 10^{-3}$). The gray dots represent group 1, and the black dots represent group 2. Only probe placements with at least one valid site are shown. **k** Relationship between the contribution of the recurrent component and the cutoff SF in L2/3 (Pearson's correlation, $n = 53$, $r = 0.4$, $p = 0.003$). Gray lines in (**g**, **j**, **k**) represent linear regressions which calculated for correlation measurements. Data are presented as mean values ± SEM for (**e**, **f**, **h**, **i**). *$p < 0.05$, **$p < 0.01$, ***$p < 0.001$. Source data are provided as a Source Data file.

mechanism) and the two groups, we plotted the contributions of the three components across different cortical layers (Fig. 6h) for the two groups. The contributions of the three components were significantly different between the two groups in the output layer (L2/3). The contribution of the M pathway was significantly greater in group 1 (Fig. 6h; $n = 62$, $0.39 \pm 0.03$ for group 1; $n = 109$, $0.25 \pm 0.02$ for group 2; $p < 10^{-4}$, rank-sum test), and the contribution of the P pathway was significantly greater in group 2 (Fig. 6h; $n = 62$, $0.38 \pm 0.03$ for group 1; $n = 109$, $0.47 \pm 0.02$ for group 2; $p = 0.006$, rank-sum test). The contribution of the recurrent component to group 2 was significantly greater than that to group 1 (Fig. 6h; $n = 62$, $0.23 \pm 0.01$ for group 1; $n = 109$, $0.28 \pm 0.01$ for group 2; $p = 0.03$, rank-sum test). Furthermore, the contribution of the recurrent mechanism was significantly greater in group 2 at high SFs (Fig. 6i; $p < 0.001$ at SFs higher than 8 cycles/degree, rank-sum test) and was significantly correlated with output/input activation (Fig. 6j; $n = 53$, $r = 0.5$, $p < 10^{-3}$). In summary, as the output layer was dominated by the P pathway in group 2 (Fig. 6h), the

contribution of the recurrent component was also significantly greater in group 2 at high SFs (Fig. 6h), which suggests that the recurrent mechanism is closely related to the P pathway. The cutoff SF was significantly correlated with the contribution of the recurrent component, indicating that the recurrent mechanism plays an important role in high-SF processing (Fig. 6k; $n = 53$, $r = 0.4$, $p = 0.003$).

## Discussion

Our study provides a complete picture of the dynamics of the laminar processing mechanism for spatial frequency (SF) information in macaque V1s and reveals two types of neural mechanisms (amplification and suppression) that govern multiple laminar response patterns evoked by different SFs (Figs. 1–3). We further demonstrated that the various laminar processes were largely caused by local circuits specific to the M or P pathway (Figs. 4–6). Our main results are summarized in Fig. 7. For one group of laminar response patterns (group 1), the responses in the output layers were suppressed not only at low SFs but

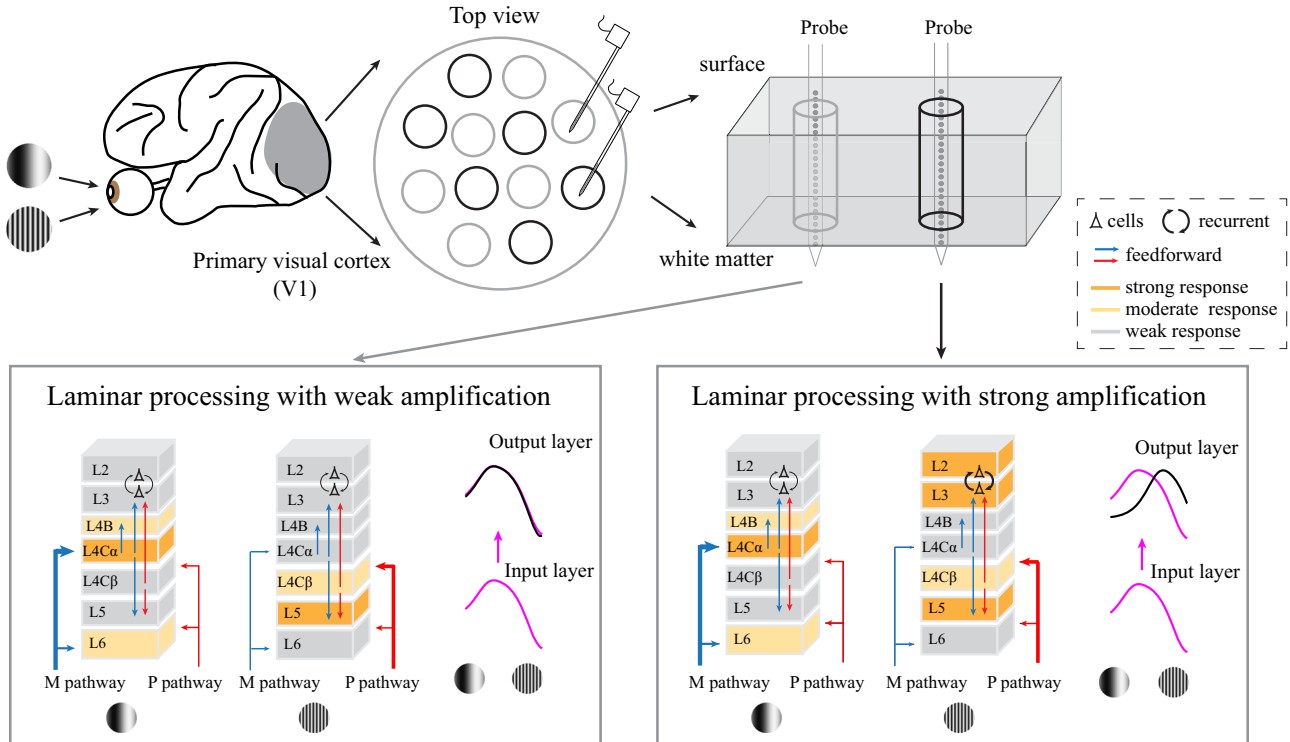

**Fig. 7 | Summary of pathway-specific laminar processing patterns for different SFs.** The upper panels illustrate our experimental paradigm. The gray circles represent group 1, and the black circles represent group 2. The lower panels illustrate the pathway-specific laminar processing patterns produced for low and high SFs. Layers with strong responses are indicated by darker orange shading. Layers with moderate responses are indicated by light orange shading. Layers with weak responses are indicated by light gray shading. The blue arrows indicate the feedforward connections of the M pathway. The red arrows indicate the feedforward connections of the P pathway. The black arrows in L2 and L3 indicate local recurrent connections. The thickness of an arrow represents the strength of its corresponding connection. The purple curve indicates the SF tuning process inherited from the input layer. The black curve indicates the SF tuning process performed for the output layer.

also at high SFs (cross-layer suppression). For the other group of laminar response patterns (group 2), the responses were amplified in the output layers at high SFs (cross-layer amplification) but suppressed at low SFs. The selective amplification of high-SF responses led to an increase in the SF preference in the V1 output layers, which enhanced the representation of high-SF information. The variety exhibited by laminar processing was mostly governed by the local circuits within the columns driven by the M and P pathways. Cross-layer suppression was largely caused by the feedforward mechanisms in the M and P pathways, while the cross-layer amplification of high-SF responses was due to the local recurrent mechanism, which predominantly occurred in the P pathway. Taken together, our results suggest that the laminar processing mechanism in the primate V1 is not uniform but is M or P pathway-specific. Pathway-specific laminar processing originates in the local circuitries within columns and selectively improves the cortical responses to high spatial frequencies.

Although the anatomical structures of the M and P pathways in V1 have been well documented[1,4,7], the neural computations of the two pathways in V1 remain unclear. One contribution of our study is that we clearly described the activation patterns for the two pathways by taking advantage of their different spatial frequency preferences. Based on the distribution of the cutoff SFs for the V1 layers (Fig. 3c), we found that the two input layers (L4Cα and L4Cβ) had significantly different cutoff SFs corresponding to the SF preferences of the M and P pathways (Fig. 3c, d). L2/3 and L5 had high cutoff SFs comparable to the cutoff SFs of L4Cβ (Fig. 3c), which suggests that they have a strong correlation with the subnetwork for the P pathway. The cutoff SFs for L4Cα, L4B and L6 were similar (Fig. 3c), and all were lower than that of L4Cβ, suggesting that these V1 layers are closely related to the

subnetwork for the M pathway. The contributions of the M and P pathways to each V1 layer, estimated by a computational model (Fig. 6d), were consistent with the SF preference results. The V1 subnetwork for the M pathway strongly activated L4Cα and L4B, and it also activated neurons in L2/3 through a feedforward mechanism. The subnetwork for the P pathway strongly activated L4Cβ, and the activation of the subnetwork in L2/3 was governed by both feedforward and recurrent mechanisms. The neurons in the V1 output layer (L2/3) were activated by both the M and P pathways; this is consistent with studies that found that lesions in either the M or P layers of the LGN can strongly affect the response properties of V1 cells[55–57].

The overall laminar variations exhibited by the SF preferences in V1 as shown in our study (Fig. 3c, d) are consistent with previous studies on both cats[58] and macaques[30]. Tootell and colleagues[30] measured the uptake pattern of 14C-2-deoxyd-glucose (DG) induced by the gratings of high and low SFs for different V1 layers of a monkey and showed that the DG uptake was highest in L2/3 and L4Cβ at high SFs, which suggested greater SF preferences in these layers. However, it was difficult to quantitatively measure SF tuning levels and preferences because only two SF conditions were used for DG uptake in a single animal. Therefore, our study provides a more quantitative measure of the overall laminar variations induced by SF tuning and preferences in monkey V1.

In addition to the laminar variations exhibited by the SF preferences in the vertical direction within the V1 columns, we also found diverse laminar response patterns among the recording positions located in different columns along the horizontal direction of V1 (Fig. 2 and Supplementary Fig. 5). The diverse laminar response patterns induced by high SFs can be categorized into two groups (suppression

and amplification) based on the response relationships between their input and output layers. We found that the cutoff SFs for the two groups were also significantly different. The cutoff SFs in the input layers (for both M-related L4Cα and P-related L4Cβ) were similar for the two groups of V1 columns (Fig. 3c, d), but recording sites in L2/3 within one group had higher cutoff SFs than those in L4Cβ, while the recording sites in L2/3 within the other group had cutoff SFs similar to those in L4Cβ (Fig. 3c–e). This SF representation difference emerged in the V1 output layer but did not inherit the SF coding in the input layer, which suggests that laminar processing from the input layers to output layers in V1 cannot be considered a single or uniform mechanism.

We hypothesize that the two groups of laminar patterns may be associated with the anatomical structures of the blobs and interblobs in V1. Previous studies have shown that cortical columns are tuned to different SFs in cats[59] and macaques[30] and have suggested that SF preferences are different for the cortical columns located in V1 blobs and interblobs[30,44,45,60]. Tootell and colleagues[30] showed that the DG uptake induced by a high SF is greater in interblobs than in blobs. Electrophysiological studies have further confirmed that the SF preferences of neurons in interblobs are greater than those in blobs[44,45]. More recently, imaging studies (both optical imaging and two-photon imaging) revealed functional maps with columns preferred different SFs (SF maps) and suggest an alignment between blob-interblob system and SF maps[46,47,60]. Based on these previous studies, we speculated that our recording penetrations with greater SF preferences in layer 2/3 were more likely to be in/near V1 interblobs and that those with lower SF preferences in layer 2/3 were likely in/near V1 blobs. With this speculation, the results of our study are not only consistent with those of previous studies on the relationships between SF preferences and the anatomical structures of blobs and interblobs but also indicate differences between the laminar processing mechanisms for SF information in the two structures.

Our study was not able to provide direct evidence for the relationships between our recording sites and cytochrome oxidase architectures. In future work, one can first map the locations of suspected blobs and interblobs in vivo via optical imaging or two-photon imaging and then insert electrodes into specific columns (blobs or interblobs). After all the recordings are completed, the relationship between the electrode tips and cytochrome oxidase architecture can be confirmed by cytochrome oxidase (CO) histochemistry in vitro. Such future studies will provide more insights into the neural mechanisms involved in laminar processing in blobs and interblobs.

Our study assumed that the laminar processing mechanism in V1 is governed mainly by the M and P pathways. This assumption might be limited because V1 also receives inputs from the koniocellular (K) pathway, which is another important pathway relayed by the "intercalated" layers of the LGN[61]. The projection of the K pathway is segregated from those of the M and P pathways in V1. Anatomical studies have shown that the K pathway provides direct inputs to L1–3 and L4A[62–64]. Previous studies have also shown that the K pathway can directly activate the superficial layers of V1 and influence the receptive field properties of the V1 cells in L2/3[65,66]. Apart from its layer-specific organization scheme in V1, the direct K pathway input within L2/3 also has a specific pattern, which coincides perfectly with blob[63,64]. Previous studies have suggested that the K cells in the LGN prefer low spatial frequencies, and their cutoff SFs are significantly lower than those of P cells[67]. The low cutoff SF for the K pathway indicates that a response pattern with a high spatial frequency cutoff in our study may not be directly affected by the K pathway. However, we cannot exclude the possibility that the response pattern for a low spatial frequency is directly influenced by the K pathway.

To further reveal the neural mechanisms of multiple pathways (M, P and K) in V1, it is important to evaluate the response properties of all V1 sublayers, including L4A, the upper L6 and the lower L6. For example, L4A is a thin sublayer (~50 μm) that receives direct inputs

from the P and K pathways[62,64]. We did not show the results of L4A because it is technically difficult to accurately detect thin sublayers with our linear probe possessing recording channels separated by 100 μm. Possible surface cortex damage caused by the acute insertion of the probe may also have affected the accuracy of the laminar alignment process in our study. In future work, one can use electrodes with denser channels and thinner tips to precisely detect more sublayers.

What is the function of the multiple neural mechanisms (suppression and amplification) demonstrated for V1 in our study? One possible contribution of the suppressive mechanisms is to enable the brain to adapt to the statistics of its visual environment. In natural scenes, power is predominant and redundant at low spatial frequencies (SFs), and the power of high-spatial-frequency information, which is important for the surface textures and contours of objects, is relatively low[68]. Previous studies have suggested that suppressive mechanisms play important roles in balancing unevenly distributed spatial frequency powers by suppressing the neural responses obtained at low SFs[69,70]; suppressive mechanisms may also yield increased feature selectivity when the SF is low[32,34]. Indeed, we found that V1 selectively suppressed the responses to low SFs, which is consistent with previous studies.

More importantly, we found that the representation of high-SF information was enhanced by a recurrent mechanism that amplified the neural responses to high SFs, which led to greater SF preferences for more neurons in the V1 output layers than in the V1 input layers (Fig. 3c–e). To the best of our knowledge, selective amplification and better spatial resolution in the V1 output layers have not been previously reported. Selective amplification can improve the signal-to-noise ratio when the feedforward mechanism driven by high SFs is weak. Recurrent mechanisms involving selective amplification may also contribute to SF selectivity in other visual cortices, for example, functional domains encoding high SFs in V2/V4, as revealed by a recent study[60]. Moreover, the recurrent mechanism for high SFs involves slow processing with a time peak of approximately 65 ms, which is slower than that of the feedforward mechanism for low SFs (with a peak at approximately 52 ms). The slow recurrent mechanism for high SFs may be related to the coarse-to-fine processing strategy in visual perception, in which the fine details transmitted by high SFs are processed after the coarse structure of a stimulus carried by low SFs[71].

Our current study suggests that the local recurrence mechanism is P pathway-specific. Whether amplification occurs in other pathways (M and K) is an open question. We effectively activated the P pathway by using static gratings briefly presented at different orientations, and SFs may mainly activate ventral visual areas that prefer stimuli with texture and form information[72]. It is not yet known whether strong amplification of the M pathway would occur if we used stimuli with motion information (such as moving bars or gratings[14,33], or more complex motion stimuli[73]). Studies using multiple types of visual stimuli with motion, color and texture information will help us further test the origins of amplification effects in the output layers. It is also interesting to explore whether the amplification effect that enhances the responses to high SFs in the output layer over those of the input layer also exists in the V1s of other species, such as carnivores or rodents.

## Methods
### Preparation of awake monkeys

All procedures were conducted in compliance with the National Institutes of Health Guide for the Care and Use of Laboratory Animals and were approved by the Institutional Animal Care and Use Committee of Beijing Normal University. Three male adult rhesus monkeys (DQ, DK and QQ; *Macaca mulatta*, 5–7 years old, 6–9 kg) were used. This sample size is too small to make meaningful statements about the effect of sex on the findings that we found. Previous studies of primary visual cortex using macaque monkeys did not reported relevant

functional difference between male and female. Since our study focuses on primary visual cortex, differences between sexes are unlikely to our scientific findings. Under general anesthesia induced with ketamine (10 mg/kg) and maintained with isoflurane (1.5–2.0%), a titanium post was attached to the skull of each monkey with bone screws to immobilize the animal's head during behavioral training. After the animal had been trained in a simple fixation task, a circular titanium chamber (20 mm in diameter) with a removable lid was fixed over the craniotomy (15 mm anterior to the occipital ridge and 14 mm lateral from the midline) with dental cement to obtain chronic recordings from the primary visual cortex (V1). Antibiotics and analgesics were used after the surgery.

### Behavioral task

A trial began when a monkey began fixating on a 0.2° fixation point (FP) presented on a CRT screen. In each trial, the FP was displayed in the center of the screen. The animal's eye positions were sampled at 120 Hz using an infrared tracking system (ISCAN). Within 300 ms after the presentation of a FP, the animal was required to fixate on a space within an invisible circular window (with a radius of 1°) around the FP. After the animal maintained its fixation for 400 ms, the stimulus was displayed for 3 s, followed by a blank interval of 400 ms. The FP then disappeared, and the animal received a drop of water as a reward. A trial was aborted if the animal's fixation moved outside the fixation window.

### Electrophysiological recording

We simultaneously recorded the neuronal activity exhibited by different layers in V1 using a linear array (V-probe, Plexon; 24 recording channels spaced 100 µm apart, each 15 µm in diameter). The linear array was controlled by a microelectrode drive (NAN Instruments, Israel), and the depth of each probe placement was adjusted to extend through all V1 layers. To reduce the effects of cortical dimpling and cortical damage caused by the probes, after each probe penetration, we waited for at least 30 minutes before collecting data. The raw data were acquired with a 128-channel system (Blackrock Microsystems). The raw data were high-pass filtered (7th-order Butterworth filter with a 1000-Hz corner frequency), and multiunit spiking activity (MUA) was detected by applying a voltage threshold with a signal-to-noise ratio of 5.5. The raw data were also low-pass filtered (7th-order Butterworth filter with a 300-Hz corner frequency) to obtain local field potentials (LFPs). The MUA and LFPs were all downsampled to 500 Hz.

### Visual stimulation

Visual stimuli were generated with a stimulus generator (ViSaGe; Cambridge Research Systems) that was controlled by a PC running a custom-written C ++ program developed in our laboratory. The stimuli were displayed on a 22-inch CRT monitor (Dell, P1230, 1200 × 900 pixels, mean luminance 45.8 cd/m², 100 Hz refresh rate). The viewing distance was 114 cm. Two types of stimuli were used. Sparse noise was applied to simultaneously map different receptive fields (RFs). Random orientations and spatial frequencies were used to measure the dynamic responses and align the laminar positions.

### Receptive field mapping

After manually mapping the receptive fields (RFs) of the recording channels, we used sparse noise to identify the precise RF center[74]. The sparse noise consisted of a sequence of randomly positioned (usually on a 13 × 13 or 11 × 11 sample grid) dark and bright squares (0.1–0.2°, contrast = 0.9) against a gray background (luminance = 45.8 cd/m²). Each sparse noise image appeared for 20 ms, and this process was repeated at least 50 times. The sequence was divided into small segments based on the trial length. We obtained a two-dimensional map of each channel. The responses averaged from the X and Y axes of each map were fitted with a one-dimensional Gaussian function to estimate

the center position. The receptive fields were located in the parafoveal region (1–6° eccentricities).

### Dynamic responses induced at different spatial frequencies

After completing the RF mapping experiment, a sequence of random flashes for gratings with different orientations and spatial frequencies (SFs) was used to measure the dynamic responses. The sinusoidal gratings of nine different orientations (equally spaced from 0 to 180°) plus "blanks" (defined as uniform frames with the same luminance as the mean luminance of the grating images; 10% of all stimuli) were used. For each grating, the spatial phase was also varied: each grating in the set was presented at eight different spatial phases (equally spaced from 0° to 360°). Several SFs (ranging from 0.5 to 20 cycles/degree, 5–9 conditions) were used and fixed within the trial. The SF range was consistent with that used in previous studies[60,70,75]. The size of the grating was 4° to 8° and fixed within each session for large stimulus conditions ($n = 59$; $n = 16$ for DQ; $n = 26$ for DK; $n = 17$ for QQ). For sessions with small stimuli, the size of the grating ranged from 0.8° to 1.2° ($n = 32$; $n = 13$ for DQ; $n = 13$ for DK; $n = 6$ for QQ). The contrast of the grating was 90%. The gratings and blanks were randomly chosen and consisted of a sequence for each spatial frequency. Each stimulus in a sequence was randomly chosen and flashed for 20 ms with at least 50 repetitions (the number of repetitions varied from 50 to 150 between recording sessions). The sequence was divided into small segments based on the trial length (typically 3 s, with 150 stimuli). Each trial displayed one segment until all segments were used. The dynamic response at each site was smoothed with a rectangular window filter possessing a width of 20 ms (10 time points). We used the stimulus-driven energy ratio (SER) to select visually driven sites. To define the SER, we calculated the energy of all orientations at different time delays as Energy $(θ, t)$ = Resp $(θ, t)^2$ for each SF condition. We then averaged all orientations and defined the peak time as the time delay at which the energy reached its maximum value. The SER for each SF condition was then calculated as the maximum energy divided by the mean energy before the onset of a stimulus (−20–0 ms). We used the maximum SER value among all SF conditions as the final SER. MUA levels with final SER values greater than 3 were used for a further analysis.

To obtain the laminar pattern of the signal-to-noise ratio (SNR) of each SF, we first calculated the variance in the responses to the orientation at different time delays to obtain the dynamic variance. The dynamic SNR was then calculated as the dynamic variance divided by the mean variance at the baseline (−20 to 0 ms before the onset of a stimulus).

### Laminar alignment

The detailed methods used to determine laminar alignment have been previously described[34]. To align different probe placements in terms of depth, we used the laminar pattern of the MUA responses combined with a current source density (CSD) analysis of the LFP signals. The MUA and CSDs across laminar channels were measured while presenting random orientations. We averaged the responses induced under all stimulus conditions and calculated the MUA and CSD laminar patterns of every probe placement. We then summarized the common features used to guide laminar alignment. Because the thickness of the cortex and depth of the probe differed between probe placements, we assigned the recording site of each channel to a relative depth (normalized cortical depth, ranging from 0 to 1).

### SF tuning curves and cutoff SF

The SF tuning curves were measured with random presentations of stimuli possessing different orientations and spatial frequencies. Gratings were presented for 3 s with fixed SFs, and the raw response was defined as the mean firing rate during this period. Spontaneous firing rates were measured with a uniform screen possessing the same

mean luminance as that of the grating stimuli during the time period (0.4 s) before the gratings were presented. The response to each SF was calculated by subtracting the spontaneous firing rate from the raw response. The SF tuning curves were fitted by the following equation:

$$R(SF) = A_1 \times \exp\left(-\frac{SF^2}{2\sigma_1^2}\right) - A_2 \times \exp\left(-\frac{SF^2}{2\sigma_2^2}\right) \qquad (1)$$

The goodness of fit (gof) was defined as follows:

$$gof = 1 - \frac{\sum_1^n \left(R_{data}(i) - R_{fit}(i)\right)^2}{\sum_1^n \left(R_{data}(i) - \overline{R_{data}}\right)^2} \qquad (2)$$

where n is the number of SF conditions, $R_{data}$ denotes the observed response data, $R_{fit}$ is the fitting response, and $\overline{R_{data}}$ is the mean value of the observed responses. Only the sites with high goodness of fit values (larger than 0.8) were used to calculate the cutoff SF. The cutoff SF was defined as the maximal SF with response amplitudes exceeding 50% of the maximum response. The difference between the cutoff SFs of the input layer and the output layer (relative cutoff SF) was calculated with the following equation:

$$relative\ cutoff\ SF = log_{10}\left(\frac{cutoff\ SF_{output}}{cutoff\ SF_{input}}\right) \qquad (3)$$

To compare the response properties produced for low and high SFs, we selected one low SF condition and one high SF condition for each probe placement based on the SF tuning strategy of L4Cβ. The low SF condition was defined as the lowest SF among all conditions, which was typically 0.625 for most probe placements (as shown in Figs. 1h and 3g). The high SF condition was defined as the highest SF with response amplitudes greater than 90% of the maximum response (higher than or equal to 6 cycles/degree). The high SF condition for each probe placement was used for further analyses, such as histogram in Fig. 1h, K-means clustering (Supplementary Fig. 2) and correlation analysis (Figs. 3d, 4e, 6g, j, k).

### K-means clustering

We first calculated the average response patterns at each probe placement for high SFs across time (n = 59, average response during 50 ms around the peak response time; the high SF condition). For each probe placement, we sorted the channels into five cortical layers (L2, L3, L4B, L4Cα and L4Cβ) and calculated the mean response within each layer. Next, we used the matrix of all probe placements in the five cortical layers to perform k-means clustering (Supplementary Fig. 2) and calculate the response differences, as shown in Fig. 1g. We performed principal component analysis (PCA) for dimensionality reduction purposes and selected the first two PCs for visualization.

**Granger causality analysis.** Multivariate Granger causality was calculated using the MVGC MATLAB Toolbox[76]. For a time series $X$ with dimensions of [$E,T,N$], where $E$ is the number of electrodes, $T$ is the trial length and $N$ is the number of trials, we first estimated the corresponding VAR model parameters [$A_k,\Sigma$] (tsdata_to_var.m) with a fixed order of 15 (corresponding to 30 ms; similar results can be obtained using the optimal order determined based on information criteria). We then calculated the autocovariance sequence $\Gamma_k$ (var_to_autocov.m) from the VAR parameters. Finally, we obtained the time-domain conditional Granger causality value (autocov_to_pwcgc.m) with dimensions of [$E,E$], where the first dimension indicates the receiver of the projection (to) and the second dimension indicates the sender of the projection (from).

**Model fitting and evaluation.** To dissect the feedforward and recurrent components underlying the dynamic responses, we fit a three-component model to the dynamic responses of each recorded channel. The parameters were determined by the 'fmincon' MATLAB function. We fitted the neuronal responses evoked by different SFs at a V1 layer (represented by i) as $R_i(sf,t)$ with two computational models. In the FF model (no recurrent component), the neuronal responses in the input layers of the M and P pathways (L4Cα and L4Cβ), $R_m(sf,t)$ and $R_p(sf,t)$, respectively, were linearly convolved with a temporal kernel and then weighted and summed, yielding the following neuronal responses $R_i(sf,t)$:

$$R_i(sf,t) = w_{i,m} \times R_m(sf,t)*K_{i,m}(t) + w_{i,p} \times R_p(sf,t)*K_{i,p}(t) \qquad (4)$$

where $w_{i,m}$ is the weight of the recording site due to the response in L4Cα, and $K_{i,m}(t)$ is the feedforward temporal kernel representing the transmission of a signal from L4Cα to sites in other layers with a log-normal form, as follows:

$$K_{i,m}(t) = e^{-\frac{(\log(t) - \triangle t_{i,m,1})^2}{2\sigma_{i,m,1}^2}} - g_{i,m} \times e^{-\frac{(\log(t) - \triangle t_{i,m,2})^2}{2\sigma_{i,m,2}^2}} \qquad (5)$$

$$K_{i,p}(t) = e^{-\frac{(\log(t) - \triangle t_{i,p,1})^2}{2\sigma_{i,p,1}^2}} - g_{i,p} \times e^{-\frac{(\log(t) - \triangle t_{i,p,2})^2}{2\sigma_{i,p,2}^2}} \qquad (6)$$

where $\triangle t_{i,m,1}$ and $\triangle t_{i,m,2}$ control the peak time of the temporal kernel and $\sigma_{i,m,1}$ and $\sigma_{i,m,2}$ control the width of the temporal kernel. The computations of $w_{i,p,1}, w_{i,p,2} \triangle t_{i,p,1}, \triangle t_{i,p,2}, \sigma_{i,p,1} \sigma_{i,p,2}$ and $g_{i,p}$ are similar to those of $w_{i,m,1}, w_{i,m,2}, \triangle t_{i,m,1}, \triangle t_{i,m,2}, \sigma_{i,m,1}, \sigma_{i,m,2}$ and $g_{i,m}$ but for the P pathway.

In the FF & Rec model (including a recurrent component), in addition to feedforward convergence from the input layer, the sites in the output layer received an SF-dependent component:

$$R_i(sf,t) = w_{i,m} \times R_m(sf,t)*K_{i,m}(t) + w_{i,p} \times R_p(sf,t)*K_{i,p}(t) + w_{i,rec}(sf) \times R_{i,rec}(t) \qquad (7)$$

where $w_{i,rec}(sf)$ represents the weight of the recurrent component, which was dependent on sf, and $R_{i,rec}(t)$ represents the dynamic response of the recurrent component with a log-normal form:

$$R_{i,rec}(t) = e^{-\frac{(\log(t) - \triangle t_{i,rec})^2}{2\sigma_{i,rec}^2}} \qquad (8)$$

where $\triangle t_{i,rec}$ and $\sigma_{i,rec}$ control the peak time and width of the dynamic responses, respectively.

The raw goodness of fit (gof) was calculated with the following equation:

$$gof_{raw} = 1 - \frac{\sum_{i=1}^n \sum_{t=0}^{150} \left(R_{data}(i,t) - R_{model}(i,t)\right)^2}{\sum_{i=1}^n \sum_{t=0}^{150} \left(R_{data}(i,t) - \overline{R_{data}}\right)^2} \qquad (9)$$

To evaluate the goodness of fit of the FF model and FF & Rec model with different numbers of parameters, we defined the adjusted gof as described by the following equation:

$$gof = 1 - \frac{(N-1)(1 - gof_{raw})}{N - P - 1} \qquad (10)$$

where N is the number of samples (combined time and SF domain) and p is the number of variables. Only the sites with high goodness of fit values (larger than 0.8) were used to analyze the relative contributions of the three components.

## Contributions of different mechanisms

The contributions of the three mechanisms to the output laminar activation patterns were calculated for different SFs (Fig. 6e–g). We first obtained the laminar activation patterns (LAPs) of the three mechanisms from the FF & Rec model for each probe placement. The corresponding functions are as follows:

$$LAP_m(sf,t,i) = w_{i,m} \times R_m(sf,t,i) * K_{i,m}(t,i) \tag{11}$$

$$LAP_p(sf,t,i) = w_{i,p} \times R_p(sf,t,i) * K_{i,p}(t,i) \tag{12}$$

$$LAP_{rec}(sf,t,i) = w_{i,rec}(sf) \times R_{rec}(t,i) \tag{13}$$

where i is the number of recording sites for each probe placement. Then, we defined the contribution of the M pathway as Contribution$_m$, the contribution of the P pathway as Contribution$_p$, and the contribution of the recurrent mechanism as Contribution$_{rec}$. The corresponding functions are as follows:

$$Contribution_m(sf) = 1 - \frac{\sum_{t=0}^{150}\sum_{i=1}^{n}\left(LAP_{data}(sf,t,i) - LAP_m(sf,t,i)\right)^2}{\sum_{t=0}^{150}\sum_{i=1}^{n}\left(LAP_{data}(sf,t,i) - \overline{LAP_{data}}\right)^2} \tag{14}$$

$$Contribution_p(sf) = 1 - \frac{\sum_{t=0}^{150}\sum_{i=1}^{n}\left(LAP_{data}(sf,t,i) - LAP_p(sf,t,i)\right)^2}{\sum_{t=0}^{150}\sum_{i=1}^{n}\left(LAP_{data}(sf,t,i) - \overline{LAP_{data}}\right)^2} \tag{15}$$

$$Contribution_{rec}(sf) = 1 - \frac{\sum_{t=0}^{150}\sum_{i=1}^{n}\left(LAP_{data}(sf,t,i) - LAP_{rec}(sf,t,i)\right)^2}{\sum_{t=0}^{150}\sum_{i=1}^{n}\left(LAP_{data}(sf,t,i) - \overline{LAP_{data}}\right)^2} \tag{16}$$

The contribution of each mechanism to the laminar activation patterns was further normalized by dividing by the total contribution of the three mechanisms. We calculated the relative contribution of the feedforward components of the M and P pathways as MP index = (Contribution$_p$- Contribution$_m$)/(Contribution$_p$+ Contribution$_m$).

We also calculated the contributions of the three mechanisms to the response at each recording site (Fig. 6h–k). The dynamic strength of each component at each site is reflected by $LAP_m(sf,t,i)$, $LAP_p(sf,t,i)$ or $LAP_{rec}(sf,t,i)$. For site i under stimulus condition sf, the maximum dynamic strength value was defined as the raw contribution of each mechanism. These raw contributions were further normalized by dividing by the total contribution of the three mechanisms.

## Statistical analysis

To identify significant differences, we used the Wilcoxon signed-rank test for paired data and the Wilcoxon rank-sum test for independent data. Pearson's correlation analyses were used to test the correlations between pairs of variables. Welch's ANOVA method was used to compare the means of multiple populations. All error bars represent the mean ± standard error of the mean (SEM). All $p$ values were two-tailed. To test for distribution bimodality, we used the calibrated Hartigan dip test[77], which is a version with better test sensitivity to unimodality than the original Hartigan dip test[78].

The nonparametric test (bootstrap method) in Fig.1c was implemented by the following steps: (1) for a given probe placement, we obtained the output/input activation at a high SF (value R); (2) we randomly selected 1000 samples with replacement from the low SF distribution (each sample with value M); and (2) we counted the number (K) of samples with R < M (out of 1000 total samples). The

equivalent $p$ value for the bootstrap method was calculated as 1-K/1000. In this way, any probe placement with $p < 0.05$ was determined to have significant amplification at a high SF, and any probe placement with $p > 0.05$ was determined to have suppression at a high SF; the process was similar for a low SF.

## Reporting summary

Further information on research design is available in the Nature Portfolio Reporting Summary linked to this article.

## Data availability

The data used in the figures in this study are provided in a source data file. The dataset underlying the results can be found in https://github.com/BNUTW2023/PSLPSFV1 and deposited at https://zenodo.org/records/10890515. Source data are provided with this paper.

## Code availability

The custom MATLAB functions and scripts used to produce the results presented in this study are publicly available via GitHub: https://github.com/BNUTW2023/PSLPSFV1 and deposited at https://zenodo.org/records/10890515.

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

## Acknowledgements

This work was supported by the National Natural Science Foundation of China (grant no. 32100831 [T.W.] and grant no.32171033 [D.X.]), STI2030-Major Projects (2022ZD0204600), and fellowship of China Postdoctoral Science Foundation (grant no.2021M690435), and the Fundamental Research Funds for the Central Universities (D.X.), the 111 Project Grant (BP0719032) (D.X.).

## Author contributions

D.J.X. and T.W. designed the research. T.W., W.F.D., Y.J.W., Y.L., Y.Y., Y.G.Z., T.T.Z., X.W.S. and D.J.X. performed the research. T.W., W.F.D., G.W., L.L. and D.J.X. analyzed the data. T.W., F.D. and D.J.X. wrote the paper.

## Competing interests
The authors declare no competing interests.
