## [Peer Review File · Nature Communications]

REVIEWER COMMENTS

Reviewer #1 (Remarks to the Author):

The current study includes some interesting data on spatial frequency processing in primary visual cortex in the macaque. The results are generally important because they may clarify fundamental aspects of the spatial sensitivity in primate visual cortex., and it offers an interesting new model of spatial processing in V1. However the current results would benefit greatly from increased acknowledgement of (and integration with) relevant prior results related to the current findings.

Major concerns:

The authors divide their results into two types of cortical column: 1) those which suppress responses to all SFs, and 2) those which amplify the relative response to high SF. However, to propose a new type of column on the current evidence seems too speculative, especially with so little discussion of the history of prior proposals.

For instance, in one initial paper, Maffei and Fiorentini (1977) suggested a laminar organization of sensitivity to different SFs. More recently, the elegant studies from the Callaway group (2012, 2016) related the surface-parallel organization of SF to that of orientation and retinotopy (i.e. other columns), although they only recorded from the upper layers. The work of Tootell et al suggested cortical columns tuned to SF in cat (1981), but an organization matching the blob-interblob system in monkey (1988, 1991), which has both laminar and radial components.

A related issue is that many of the prior reports consider the variation in SF sensitivity as a gradient between low and higher SFs, rather than two distinct types of response (as in the current work). It would be ideal if the authors can more quantitatively test their assumption that there are two types, rather than a gradient.

Another aspect of the current focus is that the across-depth variation in SF sensitivity in the two types of 'columns' are then interpreted as a laminar variation originating in the magno- and parvo-cellular laminae in the LGN. However as tested here, that assumption is somewhat limited and outdated. For one thing, the current account does not acknowledge (nor integrate) the differences in inputs from the koniocellular layers of the LGN, nor the differences known in blob-versus-interblob compartments within a given layer in V1 (although the existence of blobs and interblobs is vaguely referred to in a few

sentences). Finally, the authors do not even consider some layers (e.g. layer 4B, 4A, or sublayers of layers 5 and 6 – which also contain their own blob-interblob compartments).

Minor concerns:

Line 39 and 40: The laminar processing in V1 is not “similar across species” (lines 39 and 40), nor is “the circuitry within V1 is comparable to that in other sensory cortices” (lines 424-425). Instead, the laminar organization of area V1 of the primate cortex is quite different than that in V1 of non-primate species, and quite different than that in other areas in primate cortex.

It is well known that sensitivity to spatial frequency (in both psychophysics and physiology) varies strikingly across variations in retinotopic eccentricity. As I understood it, the authors did not correct for this variation, even though (based on their figures) their sampling regions did vary across eccentricity, by a factor of at least two. Thus a “medium” SF in one probe may have been a “high” SF in another probe.

Technically, the authors take their evidence at face value, without considering the possible effects of cortical dimpling, and the cortical damage produced by the insertion of the electrodes themselves. Further, the authors did not mark (and histologically reconstruct) the locations of their electrode tips – as done by Hubel and co-workers. This is particularly relevant when considering how (and/or whether) the current results match the variations in cytochrome oxidase architecture – a question which is not addressed here. Although I do not suggest that such factors could entirely account for the differences in the current results, it would add credibility to the current paper to acknowledge these limitations, at least in the discussion section.

Line 359 mentions that the enhancement of high SFs in one type of column “improves the cortical representation of high spatial frequencies”. I believe the authors agree, but the wording here might be clarified so that readers do not conclude that the high SF content in the visual environment can be “improved” beyond the optical and biological limits of the eye itself.

Please clarify the text in lines 386-389.

Line 43: should be “comprise” instead of “compose”.

Lines 162-164: “The cross-layer amplification processing in specific columns was consistent with the nonuniform distribution of connection weights in V1 at different cortical locations (such as blob and interblob) (ref 11, 43, 44).” The sentences are unclear. None of these three references discussed the “connection weights in V1”, nor did they test the optimal spatial frequency sensitivity of the blobs and interblobs.

Throughout the paper, I did not understand the term “cross layer”. Is it intended to describe connections within a given layer, across the topography? Or does the term refer to some sort of “criss-cross” arrangement, between different layers? I was unable to find a definition of the term in the text.

Lines 438-453: “laminar processing in these two structures [blobs and inter blobs] has not yet been demonstrated.” Either I don’t agree, or don’t understand, th claim.

Reviewer #2 (Remarks to the Author):

In the current study, Wang et al., studied laminar processing of visual information in the macaque primary visual cortex, by using invasive linear arrays. The main finding is that the authors found two different types of columns, with one showing typical suppressed activity in superficial output layers (L2/3) compared to the input layer (L4), while the other type showing amplified activity in the output layers to high spatial frequency grating stimuli. The latter finding is new, which has not been reported before. Through control experiments and modeling, the authors conclude that the amplification effect is mainly due to local recurrent connectivity, instead of a more global lateral connection mechanism. The modeling also shows how a weighted-convergence of the M and P pathway onto single neurons could explain the physiological findings. Overall, the results are solid. The analyses were carefully and thoroughly applied, as usual and typical in the Xing’s lab. The paper was written well. Here I have a few comments.

1. The definition of two categories may need further verification. The key is in their Figure 1H. The distribution seems to be quite uniform for me. The two clusters from K-means (Figure 1G, PC1) is not persuasive. Instead, the authors can try some distribution statistics to examine whether it is a uniform or bimodal distribution. A statistical analysis could also be applied to examine for which data points, responses in the input and output layers are significant (using a standard of p-value less than 0.05, or 0.01, for example).

2. Related to comment 1, since the distribution is rather continuous, I would suggest to compute an index, for example, the difference or ratio of the input/output response, and use this index to plot against the later on variables, for example, recurrent vs. feedforward connectivity. Such an analysis would give us more information compared to the simple two categorizations.

3. The function implications need to be further discussed.

4. The dominance of the amplification effect in output layer in the P-pathway is expected, since this is an orientation stimuli that mainly activate the ventral visual pathway.

Dear reviewers,

We would like to express our utmost gratitude to both of you for the helpful comments and constructive suggestions. Based on your suggestions, we have made substantial changes to our manuscript. Addressing your comments has improved and clarified the findings we report in our manuscript. We hope that our responses and revision are satisfactory to you. Thank you so much!

Below we provide point-by-point responses to the comments submitted by each reviewer. The changes in the manuscript were highlighted in red color for easy identification.

REVIEWER COMMENTS

Reviewer #1 (Remarks to the Author):

The current study includes some interesting data on spatial frequency processing in primary visual cortex in the macaque. The results are generally important because they may clarify fundamental aspects of the spatial sensitivity in primate visual cortex., and it offers an interesting new model of spatial processing in V1. However the current results would benefit greatly from increased acknowledgement of (and integration with) relevant prior results related to the current findings.

Major concerns:

The authors divide their results into two types of cortical column: 1) those which suppress responses to all SFs, and 2) those which amplify the relative response to high SF. However, to propose a new type of column on the current evidence seems too speculative, especially with so little discussion of the history of prior proposals.

Reply: Thank you for the valuable comments. In the original manuscript, the claim for two types of cortical columns was mainly based on the observation of k-mean clustering result for laminar response patterns, which had no statistical test. Therefore, to propose a new type of column on the current evidence is indeed speculative. In the revision, we decided to tune down the result from k-mean clustering analysis and emphasize more on the existence of two distinct neural mechanisms (suppression and amplification) for laminar processing which led to the diverse laminar response patterns induced by stimuli at high SFs. We then provided a statistical test to show that the laminar response patterns for all probe placements had a bimodal distribution significantly different from a unimodal distribution, which supported the existence of two distinct neural mechanisms (suppression and amplification). The corresponding changes are detailed in our responses to your other comments (the related issue to SF preferences). We also added more texts in the manuscript to further discuss prior results in the history that are related to our findings. The related discussions include two main sections. In the first section, we compared our results with previous studies of laminar processing of different SFs in V1 (please see the text in lines 436 to 443). In the second section, we discussed the possible anatomical origins of multiple laminar response patterns evoked by different SFs, such as blob-interblob system (please see the text in lines to 458 to 472).

For instance, in one initial paper, Maffei and Fiorentini (1977) suggested a laminar organization of sensitivity to different SFs. More recently, the elegant studies from the Callaway group (2012, 2016) related the surface-parallel organization of SF to that of orientation and retinotopy (i.e. other columns), although they only recorded from the upper layers. The work of Tootell et al suggested cortical columns tuned to SF in cat (1981), but an organization matching the blob-interblob system in monkey (1988, 1991), which has both laminar and radial components.

Reply: We have added more discussions regarding previous results and highlight the relationship between our findings and these results (please see the text in lines 436 to 443, lines 445 to 450 and lines 458 to 472). The corresponding revisions are briefly summarized in the

next two paragraphs.

The overall laminar variations exhibited by the SF preferences in V1 as shown in our study (Fig.3c, 3d) are consistent with previous studies on both cat (Maffei and Fiorentini, 1977) and macaque (Tootell et al., 1988). Tootell and colleagues (Tootell *et al.*, 1988) measured the uptake pattern of 14C-2-deoxyd-glucose (DG) induced by the gratings of high and low SFs for different V1 layers of a monkey and showed that the DG uptake was highest in L2/3 and L4C β at high SFs, which suggested greater SF preferences in these layers. However, it was difficult to quantitatively measure SF tuning levels and preferences because only two SF conditions were used for DG uptake in a single animal. Therefore, our study provides a more quantitative measure of the overall laminar variations induced by SF tuning and preferences in monkey V1.

In addition to the laminar variations exhibited by the SF preferences in the vertical direction within the V1 columns, we also found diverse laminar response patterns among the recording positions located in different columns along the horizontal direction of V1 (Fig. 2 and Supplementary Fig. 5). The diverse laminar response patterns induced by high SFs can be categorized into two groups (suppression and amplification) based on the response relationships between their input and output layers. We found that the cutoff SFs for the two groups were also significantly different. We hypothesize that the two groups of laminar patterns may be associated with the anatomical structures of the blobs and interblobs in V1. Previous studies have shown that cortical columns are tuned to different SFs in cat and macaque (Tootell et al., 1981; Tootell *et al.*, 1988) and have suggested that SF preferences are different for the cortical columns located in V1 blobs and interblobs (Born and Bradley, 2005; Edwards et al., 1995; Lu et al., 2018; Tootell *et al.*, 1988). Tootell and colleagues showed that the DG uptake induced by a high SF is greater in interblobs (Tootell *et al.*, 1988). Electrophysiological studies have further confirmed that the SF preferences of neurons in interblobs are greater than those in blobs (Born and Tootell, 1991; Edwards *et al.*, 1995). More recently, imaging studies (both optical imaging and two-photon imaging) revealed functional maps with columns preferred different SFs (SF maps) and suggest an alignment between blob-interblob system and SF maps (Lu *et al.*, 2018; Nauhaus et al., 2016; Nauhaus et al., 2012). Based on these previous studies, we speculated that our recording penetrations with greater SF preferences in layer 2/3 were more likely to be in/near V1 interblobs and that those with lower SF preferences in layer 2/3 were likely in/near V1 blobs. With this speculation, the results of our study are not only consistent with those of previous studies on the relationships between SF preferences and the anatomical structures of blobs and interblobs but also indicate differences between the laminar processing mechanisms for SF information in the two structures.

A related issue is that many of the prior reports consider the variation in SF sensitivity as a gradient between low and higher SFs, rather than two distinct types of response (as in the current work). It would be ideal if the authors can more quantitatively test their assumption that there are two types, rather than a gradient.

Reply: Thank you for the valuable suggestion. It should be clarified that we divided our probe placements into two groups based on their laminar response patterns, and the SF sensitivity in the two groups was significantly different. However, in the original manuscript, we did not have any statistical test to support this conclusion and we did not test the continuity of SF preferences for all recorded sites. To address your concern, we have added statistical analyses to quantitatively test whether the distributions for all cutoff SFs and response difference between input and output layers were continuous or not (Fig. R1.1). It turns out that the distribution of all cutoff SFs in V1 output layers is not significantly different from a unimodal distribution (Fig.R1.1a), which is not against the results in the literatures about the overall variation in SF sensitivity as a gradient in V1. The statistical test for unimodality was quantified by a calibrated Hartigan's dip test ($p = 0.19$; a nonparametric test to measure the unimodality of a distribution from a sample (Hartigan and Hartigan, 1985)). The unimodal distribution of cutoff SFs is consistent with the results of imaging studies, as you pointed out, which showed a continuous distribution of SF preferences between low and higher SFs in V1 (Nauhaus *et al.*, 2016; Nauhaus *et al.*, 2012).

Figure R1.1 a, Distribution of cutoff SFs for L2/3 (combined two groups). b, Distributions of cutoff SFs for L2/3 of different groups. Gray represents group 1, and black represents group 2. Average values are indicated by triangles. c, Distribution of response difference between output and input layers (combined two groups). d, Distributions of response difference between output and input layers for two different groups.

Although the overall cutoff SFs continuously distribute between low and higher SFs (Fig. R1.1a), the response difference between output and input layers has a distribution with two peaks, which is significantly different from unimodal distribution (Figure R1.1c; $n = 59$, calibrated Hartigan's dip test, $p = 0.02$). This result supports our conclusion about two different response patterns to high SF in primate V1. We also found that cutoff SFs were significantly different (Figure R1.1b; rank-sum test, $p < 10^{-8}$) between the two groups divided according to laminar response patterns. However, there is no available statistical test for k-mean clustering results in the original manuscript. Therefore, the claim for two distinct columns and the way to divide our data into two groups based on k-mean clustering result was not very rigorous. To address the concern from both you and the reviewer #2, we decided to tune down the k-mean clustering result about distinct column types, by moving the related results (k-means clustering) in the supplemental fig.2 (please see the text in lines 145 to 146). Instead, in the revision, we emphasized more on the existence of two mechanisms (suppression

and amplification) that governed the diverse laminar response patterns induced by high SFs. Different from previous version where we divided our data into two groups based on the k-mean clustering result, we divided all the probe placements into two groups based on suppression and amplification (Output/Input activation in high SF greater than 1 as amplification and less than 1 as suppression) (please see Fig.1g-i; lines 124 to 145). The results based on the newly defined groups are all similar to and consistent with results in the original manuscript and we have updated the related results in Figures 2-6. The above revision avoids the proposal of column types based on k-mean analysis and has more emphasis on different neural mechanisms (suppression and amplification) that govern the diverse laminar response patterns induced by high SFs, with statistical support for their bimodal distribution. In the revision, we also explicitly point out that SF preferences in V1 are continuously distributed, which is consistent with previous literatures (please see the text in lines 219 to 223). We hope that the revision can help us present our findings and results in a more coherent/clearer and rigorous way.

Another aspect of the current focus is that the across-depth variation in SF sensitivity in the two types of 'columns' are then interpreted as a laminar variation originating in the magno- and parvocellular laminae in the LGN. However as tested here, that assumption is somewhat limited and outdated. For one thing, the current account does not acknowledge (nor integrate) the differences in inputs from the koniocellular layers of the LGN, nor the differences known in blob-versus-interblob compartments within a given layer in V1 (although the existence of blobs and interblobs is vaguely referred to in a few sentences). Finally, the authors do not even consider some layers (e.g. layer 4B, 4A, or sublayers of layers 5 and 6 – which also contain their own blob-interblob compartments).

Reply: We agree with you that the assumption of across-depth variation in SF sensitivity only originating in the M and P pathways is limited. Although most visual information received in V1 is carried by M and P pathways, V1 also receives input from koniocellular (K) pathway, which is another important pathway relayed by the “intercalated” layers of LGN (Hendry and Reid, 2000). The projection of K pathway is segregated from those of M and P pathways in V1. Anatomical studies have shown that K pathway provide the direct input to L1–3 and L4A (Casagrande et al., 2007; Fitzpatrick et al., 1983; Hendry and Yoshioka, 1994). Previous studies also showed that K pathway can directly activate superficial layers of V1 and influence the receptive field properties of V1 cells in L2/3 (Chatterjee and Callaway, 2003; Klein et al., 2016). Apart from its layer specific organization in V1, the direct K pathway input within L2/3 also have specific pattern, which coincides perfectly with the blob (Fitzpatrick *et al.*, 1983; Hendry and Yoshioka, 1994). Previous studies suggested that K cells in LGN preferred to low spatial frequency, and with cutoff SFs significantly lower than those of P cells (White et al., 2001). The low cutoff SF for K pathway indicate that the response pattern with high cutoff spatial frequency in our manuscript may not directly driven by K pathway. However, we cannot exclude the possibility that the response pattern to low spatial frequency directly influenced by K pathway. We have added a discussion for K pathway in our revised manuscript (please see the text in lines 482 to 495).

We also agree with you that our paper did not directly address the relationship between our results and cytochrome oxidase architecture. Based on the finding of cutoff SFs were significantly different for the two groups, we speculated that the two groups may associate with the anatomical structures of blobs and interblobs in V1. We have added more discussions for blob-versus-interblob compartments in our revised manuscript (please see the text in lines 458 to 480). We further checked the response properties of other layers (L4B, L5 and L6) which also contain their own blob-interblob compartments. The cutoff SFs in the two groups are also significantly different in L4B, L5 and L6 (the cutoff SFs in the two groups are not significantly different only in L4Ca and L4Cb). Because L4A is a very thin layer in V1 (~50 μm), it is difficult to accurately detect it by our linear probe with recording channels separated by 100 μm apart. Therefore, we did not show the results for L4A. Instead, we added a discussion for this limitation in our revised manuscript (please see the text in lines 497 to 501). The results of L4B, L5 and L6 were added in the figure 3d, please see the text in lines 216 to 217 for the corresponding description.

Figure R1.2 Distributions of cutoff SFs for different column types and cortical layers, a for L4B, b for L5, c for L6. Gray represents group 1, and black represents group 2. Average values are indicated by triangles. ** $p < 0.01$, * $p < 0.05$, rank-sum test.

Minor concerns:

Line 39 and 40: The laminar processing in V1 is not “similar across species” (lines 39 and 40), nor is “the circuitry within V1 is comparable to that in other sensory cortices” (lines 424-425). Instead, the laminar organization of area V1 of the primate cortex is quite different than that in V1 of non-primate species, and quite different than that in other areas in primate cortex.

Reply: Thank you for pointing this out. We have changed the “Although laminar processing is thought to be an important and canonical process in the sensory cortex and to be similar across species” to “Although laminar processing is thought to be important in the sensory cortices of different species” (please see the text in line 39). We also removed the inappropriate descriptions “the circuitry within V1 is comparable to that in other sensory cortices” in the discussion section.

It is well known that sensitivity to spatial frequency (in both psychophysics and physiology) varies strikingly across variations in retinotopic eccentricity. As I understood it, the authors did not correct for this variation, even though (based on their figures) their sampling regions did vary across eccentricity, by a factor of at least two. Thus a “medium” SF in one probe may have been a “high” SF in another probe.

Reply: To address this concern, we did further analysis on recording sections regarding their eccentricities. Firstly, we measured the impact of eccentricity on cutoff SFs in both input and output layers (Figure R1.3a). There is a significant correlation between the eccentricity and cutoff SFs for both L4C α ($r = -0.5$, $p < 10^{-4}$) and L4C β ($r = -0.38$, $p = 0.003$), the correlation for L2/3 is also significant but relative weak ($r = -0.29$, $p = 0.025$). Then, we selected four SF conditions based on cutoff SF in L4C β for each probe placement (0.25, 0.5, 1 and 2 octaves, relative to cutoff SF in L4C β ; two examples in Figure R1.3b). Because the four SF conditions were based on the cutoff SF in L4C β for each probe penetration, the effect from eccentricity was removed/minimized. We re-plotted the averaged laminar patterns for the two groups using the selected four SF conditions (Figure R3c). For group 1, the response in the output layers was suppressed not only for low SFs but also for high SFs. For group 2, the response was amplified in output layers for high SFs but suppressed for low SFs. The re-plotted response patterns for the two groups are similar to that showed in Figure 2d. We have added the results of Figure R1.3 to supplemental fig.6, and added the corresponding description for the method in the manuscript. Please see that in the revised manuscript (lines from 181 to 194).

Figure R1.3 **a** Relationship between the cutoff SF and eccentricity ($n = 59$ probe placements). Each dot represents data averaged from all sites at the same probe placement within each layer. Gray represents group 1, and black represents group 2. The first column denotes L4C α , the second column represents L4C β , and the third column signifies L2/3 (r is the Pearson's correlation coefficient). **b** Example recording sites of L4C β for two probe placements. The red arrows and vertical dashed lines represent SFs relative to the cutoff SF (4 levels: 0.25, 0.5, 1, 2 octaves). The red dots represent the SFs of the stimuli nearest to these 4 levels. The horizontal dashed lines represent 0.5. **c** Population-averaged laminar patterns for different SF levels. The two groups are presented separately (the upper panels show group 1, and the lower panels show group 2). The strength of an MUA response is indicated by its color. The length of the sliding window for averaging across the depth dimension was 0.1 (relative depth). The horizontal black dashed lines represent the laminar boundaries. Each pattern of the SF response was normalized by dividing it by its maximum value. **d** Cumulative probability distributions of the output/input activations for L2/3. For each probe placement, the responses of the input layers were averaged across all sites of L4C α and L4C β . The responses were averaged from 0 to 120 ms after the onset of the stimulus. For each SF condition, sites with responses higher than 0 were included (n is the number of valid sites). Gray represents group 1, and black represents group 2. *** $p < 0.001$, rank-sum test. The average values are indicated by triangles.

Technically, the authors take their evidence at face value, without considering the possible effects of cortical dimpling, and the cortical damage produced by the insertion of the electrodes themselves. Further, the authors did not mark (and histologically reconstruct) the locations of their electrode tips – as done by Hubel and co-workers. This is particularly relevant when considering how (and/or whether) the current results match the variations in cytochrome oxidase architecture – a question which is not addressed here. Although I do not suggest that such factors could entirely account for the differences in the current results, it would add credibility to the current paper to acknowledge these limitations, at least in the discussion section.

Reply: Thank you for the suggestion. To reduce the effect of cortical dimpling and cortical damage by the probes, in each probe penetration, we waited for at least 30 minutes before data collection. Empirically, this step allowed us to get better signals and seemed to allow the tissue around the probe to be stabilized. We have added this technically details in method section (please see the text in lines 568 to 569). Because probe penetrations in the two groups both had high signal quality at low SF conditions and they were not significantly different at low SF conditions (Supplementary Fig.4a and 4e), we concluded that the neurons in the recording sites from the two groups were both effectively activated by visual stimuli. However, we indeed cannot completely exclude possible damages to the surface cortex by the acute insertion of probe. We have added statements about this limitation in discussion section (please see the text in lines 502 to 504).

We also agree with you that we were not able to directly show the relationship between our electrode tips and cytochrome oxidase architecture. In the future work, one can firstly map the locations of suspected blob and interblob in vivo by optical imaging or two-photon imaging and then insert electrodes to specific columns (blob or interblob). After finishing all recordings, one can further confirm the relationship between the electrode tips and cytochrome oxidase architecture by cytochrome oxidase (CO) histochemistry in vitro. Such future studies might provide more insights into neural mechanisms for laminar processing in blobs and interblobs. We have added more text related to the above limitations and future works in discussion section (please see the text in lines 474 to 480).

Line 359 mentions that the enhancement of high SFs in one type of column “improves the cortical representation of high spatial frequencies”. I believe the authors agree, but the wording here might be clarified so that readers do not conclude that the high SF content in the visual environment can be “improved” beyond the optical and biological limits of the eye itself.

Reply: Thank you for pointing out our unclear statement. We have replaced the statement by “Pathway-specific laminar processing originates in local circuitries within columns and selectively enhanced the cortical response to high spatial frequencies.” (please see the text in lines 414 to 416).

Please clarify the text in lines 386-389.

Reply: In this paragraph, we meant to discuss the species difference for laminar processing of

SF information. The key question is whether the enhanced response to high SFs in output layer compared with input layer also exist in other species. We have revised this text to make this point clearer (please see the text in lines 538 to 540).

Line 43: should be “comprise” instead of “compose”.

Reply: We have changed the phrase " compose " to " comprise " in the manuscript (please see the text in line 43).

Lines 162-164: “The cross-layer amplification processing in specific columns was consistent with the nonuniform distribution of connection weights in V1 at different cortical locations (such as blob and interblob) (ref 11, 43, 44).” The sentences are unclear. None of these three references discussed the “connection weights in V1”, nor did they test the optimal spatial frequency sensitivity of the blobs and interblobs.

Reply: Thank you for pointing out our unclear statement. In these sentences, we meant to point out the distinct laminar response patterns at different cortical locations in our results is consistent with the different functional columns in V1 which showed in previous studies, such as blob and interblob. We have reorganized this paragraph in the revised manuscript (please see the text in lines 200 to 203).

Throughout the paper, I did not understand the term “cross layer”. Is it intended to describe connections within a given layer, across the topography? Or does the term refer to some sort of “criss-cross” arrangement, between different layers? I was unable to find a definition of the term in the text.

Reply: The “cross layer” in our manuscript described the change of response strength/property from input layer to output layer (between L2/3 and L4C). If L2/3 have stronger response relative to L4C, we defined the response property as “cross-layer amplification”. If L2/3 have weaker response relative to L4C, we defined the response property as “cross-layer suppression”. We have added these definitions in the manuscript (please see the text in lines 121 to 124).

Lines 438-453: “laminar processing in these two structures [blobs and inter blobs] has not yet been demonstrated.” Either I don’t agree, or don’t understand, the claim.

Reply: Thank you for pointing out our unclear statement. We have deleted the text “laminar processing in these two structures [blobs and inter blobs] has not yet been demonstrated.” in the revised manuscript and reorganized the corresponding paragraph based on your other comments about “blobs and interblobs” (please see the text in lines 458 to 472).

Reviewer #2 (Remarks to the Author):

In the current study, Wang et al., studied laminar processing of visual information in the macaque primary visual cortex, by using invasive linear arrays. The main finding is that the authors found two different types of columns, with one showing typical suppressed activity in superficial output layers (L2/3) compared to the input layer (L4), while the other type showing amplified activity in the output layers to high spatial frequency grating stimuli. The latter finding is new, which has not been reported before. Through control experiments and modeling, the authors conclude that the amplification effect is mainly due to local recurrent connectivity, instead of a more global lateral connection mechanism. The modeling also shows how a weighted-convergence of the M and P pathway onto single neurons could explain the physiological findings. Overall, the results are solid. The analyses were carefully and thoroughly applied, as usual and typical in the Xing's lab. The paper was written well. Here I have a few comments.

1. The definition of two categories may need further verification. The key is in their Figure 1H. The distribution seems to be quite uniform for me. The two clusters from K-means (Figure 1G, PC1) is not persuasive. Instead, the authors can try some distribution statistics to examine whether it is a uniform or bimodal distribution. A statistical analysis could also be applied to examine for which data points, responses in the input and output layers are significant (using a standard of p-value less than 0.05, or 0.01, for example).

Reply: Thank you for the valuable suggestion. We did a statistical analysis (The Dip Test of unimodality; (Hartigan and Hartigan, 1985)) to test whether the distribution of response difference between input and output layers, shown in Figure R2.1a (replot of Figure 1H in the original manuscript), is significantly different from a unimodal distribution as follows. We combined the response difference for the two groups of probe placements (Fig. R2.1a) into one distribution for all probe placements (Figure R2.1b; $n = 59$ probe placements). The distribution for combined data has two peaks and significantly different from unimodal distribution (calibrated Hartigan's dip test, $p = 0.02$). The bimodal distribution property suggests two different response patterns to high SF in primate V1.

Figure R2.1 a, Distributions of response difference between output and input layers for two different groups. Gray represents group 1, and black represents group 2. b, Distribution of response difference between output and input layers (combined two groups).

Despite the two groups of response patterns can be clustered by K-means clustering (supplemental figure 2), we agree with you that the definition of two categories needs carefulness. Based on the concern from both you and reviewer #1, we decided to tune down the claim about column clusters in primate V1 based on the k-means analysis. In the revised

manuscript, we emphasized more on the finding of two distinct neural mechanisms (suppression and amplification) for laminar process, which led to diverse laminar response patterns induced by stimuli at high SFs (please see the text in lines 106 to 108 and lines 111 to 119). To better quantify the distribution of laminar response patterns, following your suggestion, we defined an index of Output/Input activation as the ratio between the responses of the output layer (L2/3) and input layer (L4C) for each probe placement. When the index is less than 1 for a probe placement, it indicates cross-layer suppression dominate the laminar response pattern induced by high SF; and an index larger than 1 represents cross-layer amplification dominate the laminar response pattern (Fig. R2.2a). Most probe placements showed cross-layer suppression at low SFs (57/59), but both cross-layer suppression and amplification were widely observed at high SFs (Fig. R2.2b). The Output/Input activation at high SF is significantly higher than low SF (Fig. R2.2c; $p < 10^{-6}$; signed-rank test).

Figure R2.2 a, Schematic of the quantification of laminar processing. Output/input activation was defined by the ratio of response between output layers and input layers in recording sites. The width of each trapezoid represents the response strength. b, Distributions of Output/input activation for low SF (upper panel) and high SF (lower panel). c, Relationship between the output/input activation calculated from low SF and high SF condition. Gray represents group 1, and black represents group 2. Different shapes represent the three animals (DQ: circles; DK: triangles; QQ: squares).

To further test which data points (probe placements) has significant amplification, we compared the Output/Input activation of each probe placement at high SF with the distribution of Output/Input activation at low SF (upper panel of Fig. R2.2b). The non-parametric test (Bootstrap method) was implemented by the following steps: (1) For a given probe placement, we can get the Output/Input activation at high SF (value R); (2) We randomly selected 1000 samples with replacement from distribution of low SF (each sample with value M); (3) We counted the number (K) in 1000 samples with $R < M$. The equivalent p-value for the bootstrap method is calculated as $1-K/1000$. In this way, any probe placement with $p < 0.05$ is determined to have significant amplification at high SF and any probe placement with $p > 0.05$ is determined to have suppression at high SF similar to at low SF. Among all probe placements, 29 penetrations have Output/Input activation at high SF significantly higher than 1 and the mean suppression at low SF ($n = 29$; $p < 0.05$; black dots in R2.2c) and 30 probe placements have Output/Input activation at high SF not significantly larger than 1 and mean suppression at low SF ($n = 30$; $p > 0.05$; gray dots in R2.2c). We added the results of Figure R2.2 into the new Figure 1 to summary the diverse response pattern in

high SF.

For the easy organization of our articles and the easy comparison of different response patterns (suppression and amplification), we kept our data in the form of two groups in the revised manuscript. But different from the previous version, we directly divided all the probe placements into two groups based on Output/Input activation in high SF (please see the text in lines 124 to 145), instead of based on the k-means analysis. Probe placements with Output/Input activation less than 1 were defined as group 1 (n = 30; black dots in R2.2c), probe placements with Output/Input activation higher than 1 were defined as group 2 (n = 29; black dots in R2.2c). Using the two newly divided groups, we replot results in Figure 2-6. The results in the revised manuscript are similar to and consistent with those in the old version which based on k-means clustering, but they have a better/clearer definition for the grouping. We also moved the results of k-means clustering and related results to the supplemental figure 2 (please see the text in lines 145 to 146).

2. Related to comment 1, since the distribution is rather continuous, I would suggest to compute an index, for example, the difference or ratio of the input/output response, and use this index to plot against the later on variables, for example, recurrent vs. feedforward connectivity. Such an analysis would give us more information compared to the simple two categorizations.

Reply: Thank you for the suggestion. We compute the response ratio between the output and input layers (Output/Input activation) as an index for each probe placement at high SF condition (n = 59 for large stimulus; n = 32 for small stimulus). The Output/Input activation is significant correlated between small and large stimulus condition (Figure R2.3a; $r = 0.85$, $p < 10^{-9}$), which is consistent with results for individual recording sites in Figure 4e. We also use this index to correlate other parameters (Figure R2.3b-d). The normalized GC value for high SF condition is significant correlated with Output/Input activation (Figure R2.3b; $r = 0.43$, $p = 0.01$; for small stimulus condition, n = 32). The contribution of recurrence for high SF condition in Fig6i is significant correlated with Output/Input activation (Figure R2.3c; $r = 0.5$, $p < 10^{-3}$; for small stimulus condition, n = 53). The cutoff SFs of L2/3 in Fig3e is also significant correlated with Output/Input activation at high SF (Figure R2.2d; $r = 0.38$, $p = 0.005$; for large stimulus condition, n = 59). The significant correlation between Output/Input activation at high SFs and other parameters indicates that neural mechanisms (suppression and amplification) not only lead to diverse laminar response patterns in V1, but also strongly affect functional properties in V1 output layers. We have added figure R2.2b and R2.2c to figure 5 and figure 6, respectively (please see the text in lines 324 to 327 and lines 391 to 392). We also added figure R2.2a and R2.2d to supplemental fig. 7 (please see the text in lines 232 to 236 and lines 279 to 280).

Figure R2.3 a, Relationship between the output/input activation calculated from small and large stimuli at the high SF condition. b, Relationship between the output/input activation and normalized GC value calculated from stimuli at the high SF condition. c, Relationship between the output/input activation and contribution of the recurrent component calculated from stimuli at the high SF condition. d, Relationship between the output/input activation and cutoff SFs. Gray dots represent group 1, and black dots represent group 2. r is the Pearson's correlation.

3. The function implications need to be further discussed.

Reply: We added more discussions about the function of multiple neural mechanisms (suppression and amplification) for laminar processing of spatial frequency in our study (please see the text in lines 506 to 507 and lines 521 to 529). The corresponding revisions are briefly summarized in the next two paragraphs.

“What is the function of the multiple neural mechanisms (suppression and amplification) demonstrated for V1 in our study? One possible contribution of suppressive mechanisms is to enable the brain to adapt to the statistics of the visual environment. In natural scenes, power is predominant and redundant at low spatial frequencies (SFs), and the power of high-spatial-frequency information, which is important for the surface textures and contours of objects, is relatively low (Simoncelli and Olshausen, 2001). Previous studies have suggested that suppressive mechanisms play important roles in balancing unevenly distributed spatial frequency powers by suppressing the neural responses obtained at low SFs (Bredfeldt and Ringach, 2002; Skyberg et al., 2022), suppressive mechanisms may also yield increased feature selectivity when the SF is low. Indeed, we found that V1 selectively suppressed the response to low SFs which is consistent with previous studies.

More importantly, we found that the representation of high-SF information was enhanced by a recurrent mechanism that amplified the neural responses to high SFs, which led to greater SF preferences for more neurons in the V1 output layers than in the V1 input layers (Fig. 3c-e). To the best of our knowledge, selective amplification and better spatial resolution in the V1 output layers have not been previously reported. Selective amplification can improve the

signal-to-noise ratio when the feedforward mechanism driven by high SFs is weak. Recurrent mechanisms involving selective amplification may also contribute to SF selectivity in other visual cortices, for example, functional domains encoding high SFs in V2/V4, as revealed by a recent study (Lu *et al.*, 2018). Moreover, the recurrent mechanism for high SFs involves slow processing with a time peak of approximately 65 ms, which is slower than that of the feedforward mechanism for low SFs (with a peak at approximately 52 ms). The slow recurrent mechanism for high SFs may be related to the coarse-to-fine processing strategy in visual perception, in which the fine details transmitted by high SFs are processed after the coarse structure of a stimulus carried by low SFs (Hegde, 2008).”

4. The dominance of the amplification effect in output layer in the P-pathway is expected, since this is an orientation stimuli that mainly activate the ventral visual pathway.

Reply: We agree with you that dominance of the amplification effect in output layer in the P-pathway is expected in our results. You have raised an interesting question that whether amplification occurs in other pathways (M and K). We effectively activated the P pathway by using static gratings briefly presented at different orientations, and SFs may mainly activate ventral visual areas that prefer stimuli with texture and form information (Pasupathy *et al.*, 2019). It is not yet known whether strong amplification of the M pathway would occur if we used stimuli with motion information (such as moving bars or gratings (Gur *et al.*, 2005; Hawken *et al.*, 1988), or more complex motion stimuli (Luo *et al.*, 2019)). Studies using multiple types of visual stimuli with motion, color and texture information will help us further test the origins of amplification effects in the output layers. We have added relevant discussions in the revised manuscript (please see the text in lines 531 to 538).

References

- Born, R.T., and Bradley, D.C. (2005). Structure and function of visual area MT. *Annu Rev Neurosci* 28, 157-189. 10.1146/annurev.neuro.26.041002.131052.
- Born, R.T., and Tootell, R.B.H. (1991). Spatial-Frequency Tuning of Single Units in Macaque Supragranular Striate Cortex. *P Natl Acad Sci USA* 88, 7066-7070.
- Bredfeldt, C.E., and Ringach, D.L. (2002). Dynamics of spatial frequency tuning in macaque V1. *J Neurosci* 22, 1976-1984.
- Casagrande, V.A., Yazar, F., Jones, K.D., and Ding, Y. (2007). The morphology of the koniocellular axon pathway in the macaque monkey. *Cereb Cortex* 17, 2334-2345. 10.1093/cercor/bhl142.
- Chatterjee, S., and Callaway, E.M. (2003). Parallel colour-opponent pathways to primary visual cortex. *Nature* 426, 668-671. 10.1038/nature02167.
- Edwards, D.P., Purpura, K.P., and Kaplan, E. (1995). Contrast Sensitivity and Spatial-Frequency Response of Primate Cortical-Neurons in and around the Cytochrome-Oxidase Blobs. *Vision Research* 35, 1501-1523.
- Fitzpatrick, D., Itoh, K., and Diamond, I.T. (1983). The laminar organization of the lateral geniculate body and the striate cortex in the squirrel monkey (*Saimiri sciureus*). *J Neurosci* 3, 673-702. 10.1523/JNEUROSCI.03-04-00673.1983.

Gur, M., Kagan, I., and Snodderly, D.M. (2005). Orientation and direction selectivity of neurons in V1 of alert monkeys: Functional relationships and laminar distributions. *Cereb. Cortex* *15*, 1207-1221. 10.1093/cercor/bhi003.

Hartigan, J.A., and Hartigan, P.M. (1985). The Dip Test of Unimodality. *Ann Stat* *13*, 70-84. DOI 10.1214/aos/1176346577.

Hawken, M.J., Parker, A., and Lund, J. (1988). Laminar organization and contrast sensitivity of direction-selective cells in the striate cortex of the Old World monkey. *Journal of Neuroscience* *8*, 3541-3548.

Hegde, J. (2008). Time course of visual perception: coarse-to-fine processing and beyond. *Prog Neurobiol* *84*, 405-439. 10.1016/j.pneurobio.2007.09.001.

Hendry, S.H., and Reid, R.C. (2000). The koniocellular pathway in primate vision. *Annu Rev Neurosci* *23*, 127-153. 10.1146/annurev.neuro.23.1.127.

Hendry, S.H., and Yoshioka, T. (1994). A neurochemically distinct third channel in the macaque dorsal lateral geniculate nucleus. *Science* *264*, 575-577. 10.1126/science.8160015.

Klein, C., Eyrard, H.C., Shapcott, K.A., Haverkamp, S., Logothetis, N.K., and Schmid, M.C. (2016). Cell-Targeted Optogenetics and Electrical Microstimulation Reveal the Primate Koniocellular Projection to Supra-granular Visual Cortex. *Neuron* *90*, 143-151. 10.1016/j.neuron.2016.02.036.

Lu, Y., Yin, J., Chen, Z., Gong, H., Liu, Y., Qian, L., Li, X., Liu, R., Andolina, I.M., and Wang, W. (2018). Revealing detail along the visual hierarchy: neural clustering preserves acuity from V1 to V4. *Neuron* *98*, 417-428. e413.

Luo, J., He, K., Andolina, I.M., Li, X., Yin, J., Chen, Z., Gu, Y., and Wang, W. (2019). Going with the Flow: The Neural Mechanisms Underlying Illusions of Complex-Flow Motion. *J Neurosci* *39*, 2664-2685. 10.1523/JNEUROSCI.2112-18.2019.

Maffei, L., and Fiorentini, A. (1977). Spatial frequency rows in the striate visual cortex. *Vision Res* *17*, 257-264. 10.1016/0042-6989(77)90089-x.

Nauhaus, I., Nielsen, K.J., and Callaway, E.M. (2016). Efficient Receptive Field Tiling in Primate V1. *Neuron* *91*, 893-904. 10.1016/j.neuron.2016.07.015.

Nauhaus, I., Nielsen, K.J., Disney, A.A., and Callaway, E.M. (2012). Orthogonal micro-organization of orientation and spatial frequency in primate primary visual cortex. *Nature Neuroscience* *15*, 1683-+. 10.1038/nn.3255.

Pasupathy, A., Kim, T., and Popovkina, D.V. (2019). Object shape and surface properties are jointly encoded in mid-level ventral visual cortex. *Curr Opin Neurobiol* *58*, 199-208. 10.1016/j.conb.2019.09.009.

Simoncelli, E.P., and Olshausen, B.A. (2001). Natural image statistics and neural representation. *Annual review of neuroscience* *24*, 1193-1216.

Skyberg, R., Tanabe, S., Chen, H., and Cang, J. (2022). Coarse-to-fine processing drives the efficient coding of natural scenes in mouse visual cortex. *Cell reports* *38*.

Tootell, R.B., Silverman, M.S., and De Valois, R.L. (1981). Spatial frequency columns in primary visual cortex. *Science* *214*, 813-815. 10.1126/science.7292014.

Tootell, R.B.H., Silverman, M.S., Hamilton, S.L., Switkes, E., and Devalois, R.L. (1988). Functional-Anatomy of Macaque Striate Cortex .5. Spatial-Frequency. *Journal of Neuroscience* *8*, 1610-1624.

White, A.J., Solomon, S.G., and Martin, P.R. (2001). Spatial properties of koniocellular cells in the lateral geniculate nucleus of the marmoset *Callithrix jacchus*. *J Physiol* *533*, 519-535. 10.1111/j.1469-7793.2001.0519a.x.

REVIEWERS' COMMENTS

Reviewer #2 (Remarks to the Author):

The authors have performed additional analyses, and have addressed my concerns very well. I am happy with their response, and do not have further concerns.